# Single particle cryo-EM structure of the outer hair cell motor protein prestin

Carmen Butan[1,5], Qiang Song [1,5], Jun-Ping Bai[2], Winston J. T. Tan [1], Dhasakumar Navaratnam [1,2,3 ✉] & Joseph Santos-Sacchi[1,3,4 ✉]

The mammalian outer hair cell (OHC) protein prestin (Slc26a5) differs from other Slc26 family members due to its unique piezoelectric-like property that drives OHC electromotility, the putative mechanism for cochlear amplification. Here, we use cryo-electron microscopy to determine prestin's structure at 3.6 Å resolution. Prestin is structurally similar to the anion transporter Slc26a9. It is captured in an inward-open state which may reflect prestin's contracted state. Two well-separated transmembrane (TM) domains and two cytoplasmic sulfate transporter and anti-sigma factor antagonist (STAS) domains form a swapped dimer. The transmembrane domains consist of 14 transmembrane segments organized in two 7+7 inverted repeats, an architecture first observed in the bacterial symporter UraA. Mutation of prestin's chloride binding site removes salicylate competition with anions while retaining the prestin characteristic displacement currents (Nonlinear Capacitance), undermining the extrinsic voltage sensor hypothesis for prestin function.

[1] Department of Surgery (Otolaryngology), Yale University School of Medicine, New Haven, CT, USA. [2] Department of Neurology, Yale University School of Medicine, New Haven, CT, USA. [3] Neuroscience, Yale University School of Medicine, New Haven, CT, USA. [4] Cellular and Molecular Physiology, Yale University School of Medicine, New Haven, CT, USA. [5] These authors contributed equally: Carmen Butan, Qiang Song. ✉email: dhasakumar.navaratnam@yale.edu; joseph.santos-sacchi@yale.edu

The outer hair-cell (OHC) molecular motor prestin (Slc26a5) belongs to a diverse family of transporters that includes 10 members[1]. Unlike other members of this family, and unique to membrane proteins, prestin functions as a voltage-driven motor with rapid kinetics, likely providing cycle-by-cycle amplification of sound within the mammalian organ of Corti[2,3]. However, cycle-by-cycle amplification at frequencies higher than 50 kHz, where mammals such as bats and whales can hear, may be limited by the low-pass nature of prestin's voltage-sensor charge movement, which is a power function of frequency that is 40 dB down (1%) in magnitude at 77 kHz[4–6]. The underlying basis of prestin's electromechanical capabilities resides in its unique piezoelectric-like property that drives OHC electromotility[7–11]. For members of this diverse family, known structures are dimers with each protomer showing a common 7 + 7 inverted-repeat topology containing a core and gate domain; these proteins function variably as coupled transporters and uncoupled transporters/ion channels with a range of substrates[1,12,13]. Within the Slc26 family, prestin and pendrin (Slc26a4) are unique in showing voltage sensitivity with signature nonlinear capacitance (NLC) or equivalently, displacement currents/gating charge movements[14–16]; while pendrin lacks intrinsic electromechanical behavior[17], prestin is a minimal transporter[18–20]. Prestin's NLC can be modeled, in its most simplest form, to fit a two-state Boltzmann function[21]. Two competing ideas have been proposed to be responsible for prestin's electromotility: (1) Cl⁻ serving as the (extrinsic) voltage sensor[22] and (2) intrinsic charged residues in prestin serving as the voltage sensor with Cl⁻ acting in an allosteric manner[19,23,24]. The former posits a transporter-like movement with an arrested hemimovement of Cl⁻ in the transporter cycle acting as its voltage sensor[25,26]. With more structural information, there have been competing visions of transporter mechanisms (elevator vs. rocker) in the related proteins that share similar structural folds[27,28], although how these fit with prestin's electromechanical behavior remains speculative at best. To be sure, the lack of structural information for full-length prestin has precluded an understanding of its unique molecular motor function.

In this work, we have used single-particle cryo-electron microscopy to determine the structure of prestin from gerbil (*Meriones unguiculatus*) at subnanometer resolution that confirms Oliver's initial modeling efforts[25] and, remarkably, bears high concordance with the recently determined cryo-EM structures of Slc26a9[29,30]. Prestin forms a dimer and the cryo-EM-density map has allowed us to build a nearly complete model of the protein. In combination with electrophysiological data, our structural results suggest that the inward cytosol-facing conformation is that of prestin in the contracted state. Furthermore, mutations within prestin's now structurally confirmed anion-binding site show that the extrinsic voltage-sensor hypothesis[22] is likely incorrect[26], with the wider implication that a transporter-like mechanism driving electromotility is unlikely.

## Results

### The general architecture of full-length prestin.
We used single-particle cryo-electron microscopy to obtain the structure of detergent-solubilized prestin from gerbil (*Meriones unguiculatus*) (Fig. 1, [https://www.rcsb.org/structure/7SUN]). The protein, extracted in digitonin, and purified in the presence of GDN (glyco-diosgenin), appears to be a homogeneous oligomer when assessed with size-exclusion chromatography and by negative-stain electron microscopy (Supplementary Fig. 1A–E).

The cryo-EM images obtained from plunge-frozen specimens of solubilized prestin (Supplementary Fig. 3A) revealed clear density for the transmembrane helices (TM), the cytoplasmic domains, and the micelle belt around the protein (Supplementary Fig. 3B). A density map (Fig. 1A–C) was refined to an overall resolution of 3.6 Å according to the gold-standard Fourier shell correlation (FSC) from 111,863 particles using C2 symmetry (Supplementary Figs. 2, 3E, Supplementary Table 1). The density map displayed clear secondary structural elements and densities visible for many of the bulky side chains (Supplementary Fig. 4A-0). An analysis of the local resolution of the cryo-EM map shows that the core of the structure is better resolved than the periphery (Supplementary Fig. 3C,D), with the lowest resolution observed at the tips of the extracellular loops, the cytosolic IVS loop (variable or intervening sequence), and the cytosolic C-terminal domain, which is highly flexible. This map has allowed us to unambiguously build and refine a nearly complete model of full-length prestin (Fig. 1D–F). The cryo-EM-derived structure shows that prestin oligomerizes as a dimer, with overall dimensions of ~10 nm in diameter and ~8 nm in height. Each prestin protomer comprises 14 transmembrane helices (named TM1–TM14), a C-terminal cytosolic STAS domain, and a short cytosolic N-terminal region. The 14-transmembrane segments exhibit the same inverted 7-segment repeat organization as first observed in the crystal structure of the bacterial symporter UraA[31]. Seven of the transmembrane segments (TMs 1–4, TMs 8–11), are referred to as the "core" domain and the remaining seven helices (TMs 5–7, TMs 12–14) form the "gate" domain of the structure. Prestin dimerizes in a similar fashion to Slc26a9; it forms a stable dimer through a combined molecular interface that buries about 7712 Å². The N terminus of one protomer (residues 13–75) crosses over the STAS domain (residues 505–726) of the opposing protomer so that the interface between the STAS domain and the N terminus of the opposing subunit is the largest in the dimeric assembly and buries a surface area of approximately 1438 Å² (19% of contacts). In addition, the surface area of the STAS domain of one protomer in contact with the transmembrane domain of the other protomer is approximately 1293 Å² (representing 17% of contacts). The majority of this interface is formed by residues at the tips of TM5, TM8 and TMs 12–14 helices exposed at the intracellular surface of one protomer and the residues located on the first 3 helices of the other protomer's STAS domain. Tight STAS–STAS domain interactions facilitate prestin dimerization with a buried interface of about 1192 Å² (15% of contacts). Close contacts between residues located on the antiparallel beta-strands at the N-terminal domain of each prestin subunit also mediate the dimeric assembly and cover a surface area of about 1046 Å². This is in line with our previous biochemical evidence showing the importance of prestin's N terminus to dimerization[32]. The interface area between the TM–TM domains has a modest contribution to the dimerization (Supplementary Table 2).

Last, the protein that was used for the cryo-EM experiments contained YFP at the C terminus of prestin. We do not see densities corresponding to YFP in our reconstructions. We believe this is owing to the highly mobile nature of the attachment to the C terminus of prestin.

### Comparison with the cryo-EM structure of Slc26a9.
The cryo-EM structure of prestin closely resembles the previously reported cryo-EM structures of Slc26a9, which is a representative Slc26 family member[29,30]. In particular, the dimer of Slc26a9 from mouse in "intermediate" conformation (PDB ID, 6RTF, pair-wise $C_\alpha$ RMSDs (root-mean-square deviation): 2.985 Å overall) can be fitted into the cryo-EM-density map of prestin from gerbil reasonably well. The previously observed Slc26a9 "intermediate" conformation[29] may represent a more closed conformation when compared with the "inward-open" (open toward the cytosolic side of the membrane) conformation of Slc26a9[29]. Our prestin structure superimposes onto the "inward-open" conformation of

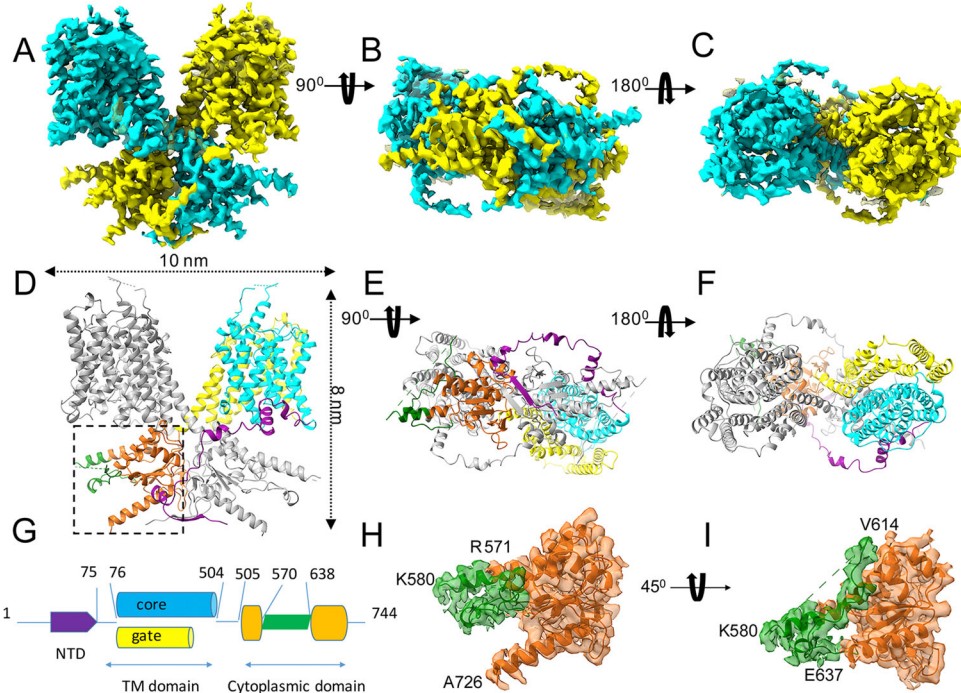

**Fig. 1 The cryo-EM structure of prestin from gerbil. A–C** Three views: side view (**A**), cytosolic view (**B**), and extracellular view (**C**) of the cryo-EM structure of the prestin dimer. The prestin structure is colored by the subunit in cyan and yellow, respectively. **D–F** Atomic model based on the cryo-EM density, shown in the same orientations as the density maps in the panels (**A–C**). The N terminus and C terminus of the prestin structure face the cytosol. The different domain structures of prestin are colored using the same color scheme as shown in the schematic representation of the prestin sequence (**G**). (**H, I**) Two close-up views, rotated ~45° with respect to each other, showing a density segment of the IVS loop overlaid with the model (shown in green ribbon) within the STAS-domain structure (shown in surface representation, in orange). Panels (**H, I**) show enlarged views of the interface boxed in panel (**D**). The positions of residues (R571, K580, and E637) from the IVS loop (shown in green ribbon) are indicated. Residue A726 is labeled to highlight the C terminus of the structure.

Slc26a9 from mouse (PDB ID, 6RTC), with pairwise $C_\alpha$ RMSDs of 4.25 Å overall. Notably, the two previously resolved "inward-open" conformations of human SLC26A9 (PDB ID, 7CH1, determined at a resolution of 2.6 Å) and mouse Slc26a9 (PDB ID, 6RTC, determined at a resolution of 3.96 Å) are essentially identical. Unless noted otherwise, all further comparisons with the "inward-open" conformation of SLc26a9 will concern the higher-resolution structure (PDB ID, 7CH1).

When aligning one prestin protomer with one Slc26a9 protomer (in "intermediate" and "inward-open" conformations) by their corresponding STAS domains (Supplementary Fig. 5A–C), the first large displacements with the Slc26a9 structures are observed at the linker connection between the transmembrane and cytosolic domains, indicated by an arrow (Supplementary Fig. 5A–C). The displacements extend into the entire transmembrane domains (Supplementary Fig. 5A–C). An intriguing result becomes apparent when superimposing the transmembrane domain of prestin onto the transmembrane domains of the Slc26a9 structures (Fig. 2A–F, Supplementary Fig. 5D–F) through the $C_\alpha$ atoms of residues within TM13 and TM14 helices. We observe differences in the orientation of TM8 (from the core domain) with respect to the TMs 13 and 14 (from the gate domain) (Fig. 2G–I; Supplementary Fig. 6A–C) at the cytosolic side of the structures. Thus, the angle between the transmembrane helices 8 and 14 is smaller by about 10° between prestin (at 57°) and the "inward-open" conformation of SLc26a9 (at 67°) and by about 4° between prestin and the "intermediate" conformation of SLc26a9 (at 61°). Consequently, the gap between the cytosolic ends of TMs 13 and 14 (from the gate domain), and the cytosolic end of TM8 (from the core domain), is smaller in prestin compared with Slc26a9, particularly in the "inward-open"

conformation (Fig. 2G–I; Fig. 2, Supplement 2A,B,C). We speculate that our cryo-EM prestin conformation, showing a narrower opening to the cytosol, would explain why this protein shows reduced ability to transport anions in comparison with Slc26a9. Moreover, the kinetic barrier afforded by this narrowing could explain the clearly measurable currents in the presence of the easily dehydrated $SCN^-$ ion that has a smaller diameter than the hydrated $Cl^-$ ion that shows reduced currents[33,34]. The position of the pair (TM3, TM10) in prestin, which contains residues important for the coordination of anion substrates, also differs from the positions of (TM3 and TM10) helices in Slc26a9. The (TM3, TM10) pair in prestin undergoes rotations relative to their counterparts in Slc26a9. Prestin's TM10 is retracted by about 12° relative to TM10 of the "inward-open" Slc26a9 structure and by about 10° relative to TM10 of the "intermediate" Slc26a9 structure (Fig. 2C, F).

When aligning the transmembrane domains of prestin and SLC26A9 structures, the $C_\alpha$ RMSD values of the superimposed structures are 4.3 Å for prestin–Slc26a9 (PDB ID, 7CH1) and 2.86 Å for prestin–Slc26a9 (PDB ID, 6RTF), respectively.

The C-terminal STAS domain of prestin is similar to the previously determined X-ray crystal structure (PDB ID, 3LLO), lacking the unstructured loop[35]. Thus, we see an identical core of five beta-sheets surrounded by 5 alpha-helices. The RMSD value when superimposing prestin's C-terminal domain solved by cryo-EM with prestin's C-terminal domain solved by X-ray crystallography (PDB ID, 3LLO) is 0.853 Å over 105 $C_\alpha$ atoms. The prestin's STAS domain and the STAS domains of the two Slc26a9 conformations align with an RMSD value of 0.955 Å over 123 $C_\alpha$ atoms (for prestin—PDB ID, 7CH1) and of 1.038 Å, respectively, over 66 $C_\alpha$ atoms (for prestin—PDB ID, 6RTF), revealing

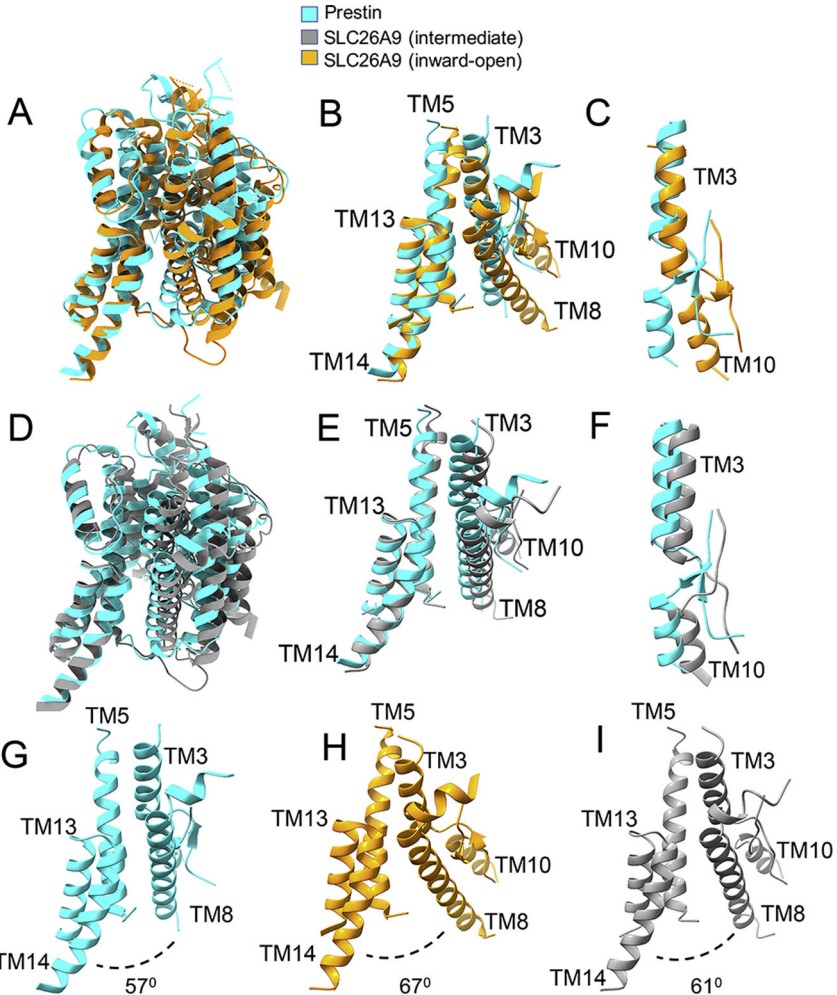

**Fig. 2 Comparison of the transmembrane regions between the prestin structure and the Slc26a9 structure (in two different conformations). A** Superposition of the transmembrane-domain structures of prestin (colored in cyan) and Slc26a9 in the "inward-open" conformation (colored in orange, PDB 7CH1). The structures have been superimposed through the TM13 and TM14 helices from the "gate" domain. **B** Close-up view showing a portion (TM3, TM5, TM8, TM10, TM13 and TM14 helices) of the overlaid transmembrane domains (as shown in panel **A**). **C** Close-up view of the overlaid pair of helices (TM3 and TM10) from the two structures in the orientations shown in panel **A**. **D** Superposition of the transmembrane domain structures of prestin (colored in cyan) and Slc26a9 in the intermediate conformation (colored in gray, PDB 6RTF) (superimposed through the gate's TM13 and TM14 helices). **E** Close-up view of the overlaid (TM3, TM5, TM8, TM10, TM13, and TM14) helices from the two structures in the orientations shown in panel (**D**). **F** Close-up view showing the offset between the pair of helices (TM3 and TM10) from the two structures in the orientations shown in panel (**D**). **G**–**I** Side-by-side comparisons of the opening between the TM8 helix (from the core domain) and the TM14 helix (from the gate domain) on the intracellular side of the membrane for the three structures. Consistent with the expectation, this opening (defined by the angle between the TM8 helix and TM14 helix) is the larger in the Slc26a9 structure (inward-open conformation) and the smaller in the prestin structure. The angle between TM8 and TM14 helices of prestin and of Slc26a9 is 57° (for prestin), 61° (for Slc26a9, the intermediate conformation), and 67° (for Slc26a9, the "inward-facing", open conformation). See also Supplementary Fig. 6.

essentially identical structures (Supplementary Fig. 5G–I). In contrast to the previously reported SLC26A9 cryo-EM structures[29,30], the cryo-EM density for the IVS loop resolved in the prestin structure shows a more ordered, interpretable density (Fig. 1H,I). Therefore, we were able to build into this more interpretable density an extended helix between residues (R571–K580) and the IVS sequence spanning residues (V614–E637). As shown in Fig. 1I, the IVS sequence (between residues 614 and 630) abuts on the tip of helix 2 (at residue G668) and the loop connecting helix3 with helix 4 (at residue N695) from the STAS domain.

**Prestin is likely in the contracted state**. We infer that the structure of prestin that we obtained is in the contracted state. Membrane depolarization from a negative resting voltage shifts

prestin from an expanded to contracted state, evoking OHC shortening[14,15]. Furthermore, increases in intracellular Cl⁻ ion concentration cause a hyperpolarizing shift in $V_h$, the voltage where, based on 2-state models, prestin is equally distributed between compact and expanded states[22,23,36]. The cell line from which we purified prestin[37] displays $V_h$ values near −110 mV in the presence of 140 mM Cl⁻ (Fig. 3A). Since the voltage across detergent micelles is effectively 0 mV, prestin is likely in a contracted state.

**Analysis of mutations in the chloride-binding site**. The analysis of mutations in the Cl⁻ binding site can provide important information. Despite purification in high chloride, we were unable to resolve a density corresponding to a chloride ion within prestin. A similar observation was made with all three structures

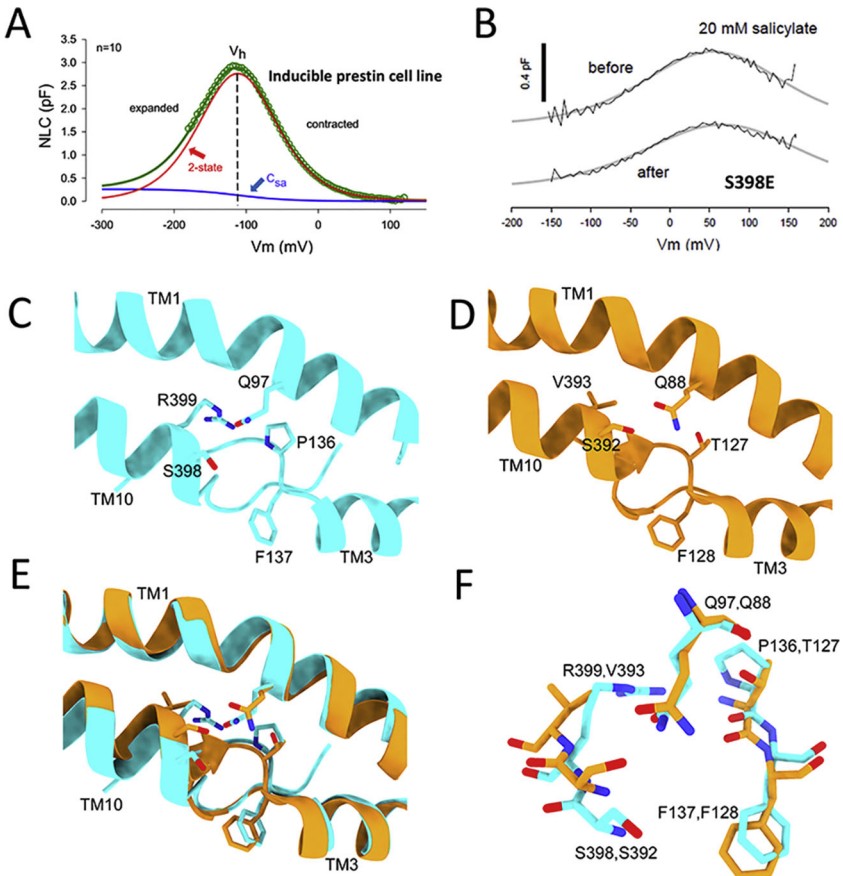

**Fig. 3 Nonlinear capacitance represents prestin voltage dependent activity and does not depend on anion binding. A** Prestin predominantly resides in the contracted state at 0 mV. Average NLC ($n = 10$), 48 h after induction of our inducible prestin HEK stable cell line (#C16). Recordings were done in the presence of 140 mM intracellular Cl. NLC-fitted (Eq. 1) parameters were Qmax = 0.40; z = 0.70; Vh = −112 mV; DCsa = 0.26 pF; NLC peaks at $V_h$; $n = 10$. **B** The S398E mutation in prestin preserves NLC after application of 20 mM salicylate. Controls shows full block of NLC. The average unitary gating charge (z) in these mutants (0.69e ± 0.03 SEM, $n = 7$) was similar to that of CHO cells expressing prestin–YFP (0.73e ± 0.14 SEM, $n = 10$, $P > 0.05$). $V_h$ was significantly different, however (96.25 mV± 4.6 WT; +73.8mV ±6.4, $p < 0.05$). **C** Close-up view showing the residues Q97, S398, R399, P136, and F137 in prestin (displayed as sticks, colored by atom type) that correspond to residues found to be important for Cl⁻ binding in Slc26a9 (with prestin in cyan ribbon). The mean z and $V_h$ values of fitted NLC of several mutations of residues that coordinate chloride binding were WT, 0.71e ± 0.03 SEM, −98.12 mV ± 2.33SEM $n = 9$; Q97A 0.65 ± 0.04, −94.1mV ± 5.29, $n = 5$; P136T 0.64 ± 0.05 SEM, −88.15 ± 10.1 SEM, $n = 9$. The differences were not significant ($p = 0.54$, one-way ANOVA for z; $p = 0.605$, one-way ANOVA for Vh). **D** Close-up view of the corresponding region in the Slc26a9 structure (the "inward-facing", open conformation, in orange ribbon). Residues Q88, S392, T127, and F128 (in stick representation) are involved in Cl-coordination in Slc26a9. **E** The location of the Cl binding site at the interface between TM1, TM3, and TM10 is conserved in prestin and Slc26a9. Overlay of portions of the transmembrane helices 1, 3, and 10, from prestin and Slc26a9, as shown in panels (**C**, **D**). **F** Overlay of the Cl-binding sites of prestin and Slc26a9 ("inward-facing", open conformation). Equivalent residues are shown as pairs. Cyan sticks are prestin residues and orange sticks are Slc26a9 residues. Source data are provided as a Source Data file.

of Slc26a9[29,30], and in prestin's case, it may be due to its relatively poor Cl⁻ binding affinity[22,38]. Nevertheless, prestin presents the canonical anion-binding site features identified in other structurally solved family members where substrates are resolved. The binding site is between TM3 and TM10 and the beta-sheets preceding these. Many of the residues important for coordinating substrate binding in those proteins are located in similar positions in prestin, including F137, S398, and R399 (Fig. 3C–F). Furthermore, with the exception of T127 in Slc26a9 (which is a proline P136 in prestin) other residues important for coordinating water molecules for substrate interactions are also conserved in prestin (Q97, F101). Although it has been speculated that Cl⁻ acts as an extrinsic voltage sensor[22], this speculation has never been confirmed; instead, anions have been shown to foster allosteric-like modulation of prestin activity and kinetics, with anion-substitute valence showing no correlation with the magnitude of prestin's unitary charge[23,24,36,38–40]. It is well known

that salicylate blocks NLC and electromotility[41,42], possibly by competitively displacing Cl⁻[22]. Here we show that mutation of S398 in the structurally identified anion-binding pocket of prestin to a negatively charged glutamate residue results in a protein that is insensitive to salicylate yet retains normal NLC (Fig. 3B). This observation is further evidence against Cl⁻ acting as an extrinsic voltage sensor. We see similar effects with R399E. Indeed, recently, Oliver et al.[43] reported on S396E that shows insensitivity to salicylate as proof refuting his original extrinsic voltage-sensor hypothesis. They also previously observed salicylate insensitivity with R399S[25]. Another mutation within the anion-binding pocket of prestin, F137 to alanine resulted in a loss of NLC or a far-right shift in its voltage sensitivity that made its detection impossible[44]; membrane insertion was confirmed by detectable SCN⁻ currents. We additionally mutated two residues in prestin that are implicated in Cl⁻ binding in Slc26a9, substituting an alanine residue for Q97, and substituting a threonine residue for P136 that

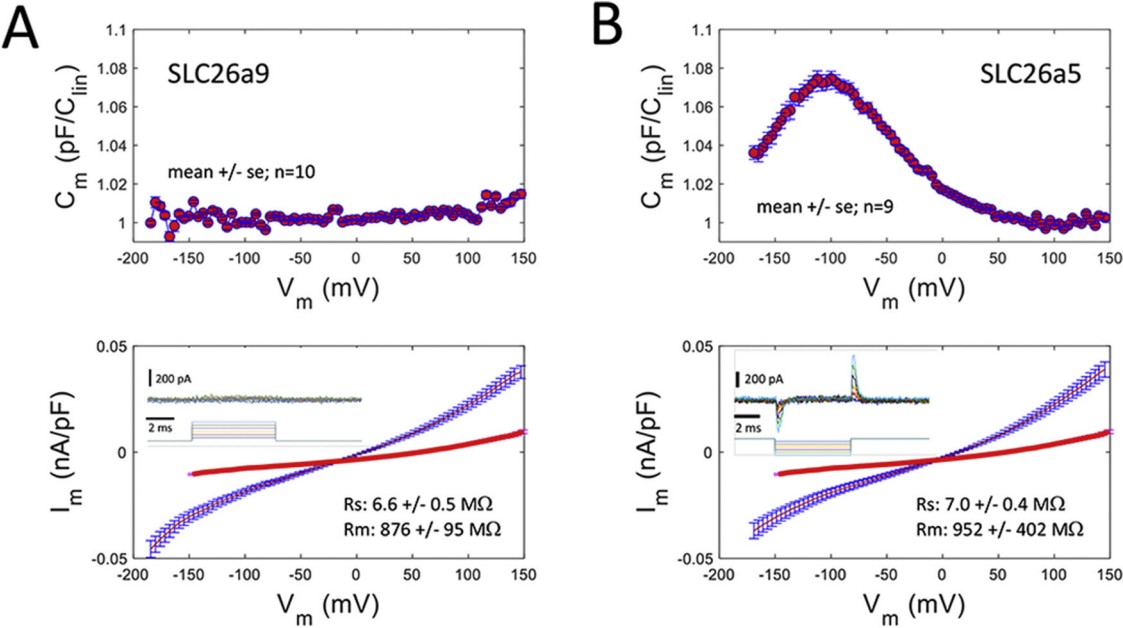

**Fig. 4 Slc26a9 is not electromechanical. A** Top panel: Capacitance of CHO cells transiently infected with Slc26a9, scaled to linear capacitance near −100 mV. Measured with dual-sine voltage superimposed on a voltage ramp from −175 to +150 mV. A very slight increase in Cm occurs at +150 mV, possibly indicating an extremely right-shifted NLC. Capacitance data are presented as mean values (± SEM) of 10 independent cells. Inset: Confocal image of CHO cells transfected with Slc26a9-YFP that is expressed on the membrane. The scale bar is 10 microns. Bottom panel: Ramp induced currents simultaneously measured with membrane capacitance, also scaled to linear capacitance. Current data are presented as mean values (± SEM) from the same 10 independent cells shown in the upper panel. Average (± SEM) series and membrane resistance are shown. In red are mean (±SEM) currents observed in untransfected cells ($n = 5$). Inset: top traces show very small nonlinear currents extracted with P/−5 protocol, subtraction holding potential set to −50 mV. Voltage protocol shown below traces. **B** Top panel: Capacitance of CHO cells transiently infected with prestin, scaled to linear capacitance near +100 mV. Measured with dual-sine voltage superimposed on a voltage ramp from −175 to +150 mV. A prominent increase in Cm occurs at −110 mV, typical of prestin NLC. Capacitance data are presented as mean values (± SEM) of nine independent cells. Bottom panel: Ramp-induced currents simultaneously measured with membrane capacitance, also scaled to linear capacitance. Current data are presented as mean values (± SEM) from the same nine independent cells shown in the upper panel. Average (± SEM) series and membrane resistance are shown. In red are mean (± SEM) currents observed in untransfected cells ($n = 5$). Inset: top traces show large nonlinear displacement currents extracted with P/-5 protocol, subtraction holding potential set to +50 mV. Voltage protocol (holding potential 0 mV) shown below traces. Source data are provided as a Source Data file.

corresponds to T127 in Slc26a9. Both these mutants have normal unitary gating charge ($z$) (Q97 0.65 ± 0.04 $n = 5$, P136 0.64 ± 0.05 $n = 9$). These z values were not significantly different ($p > 0.05$, one-way ANOVA) from WT (Fig. 3). Residue mutations within a binding pocket are well known to alter binding (e.g., affinity) in many proteins. If normal binding of anions were required for sensor charge movement, then NLC would be altered, which it is not. That these mutations do not affect NLC is evidence refuting the extrinsic voltage-sensor hypothesis, and also make implausible any associated transport-like requirements for voltage-driven electromotility.

**Slc26a9 is not electromechanical.** The marked structural similarity between prestin and Slc26a9 may provide insight into prestin's electromechanical behavior. In prestin, several charged residues have been shown to affect the size of the unitary gating charge and thus contribute to voltage sensing[19]. Of those twelve residues, nine are conserved in Slc26a9. We transiently expressed Slc26a9 in CHO cells but were unable to find NLC or gating currents in contrast to transiently transfected CHO cells expressing prestin (Fig. 4A, B top panels). Slc26a9 surface expression in transfected cells was successful as demonstrated by enhanced currents in the presence of extracellular $SCN^-$ (Supplementary Fig. 7A, B) and visualization of fluorescence on the surface of these cells expressing Slc26a9 with YFP tagged to its C-terminus (Fig. 4A, inset). Thus, despite marked structural similarities to

prestin, Slc26a9 does not mimic prestin's electromechanical behavior.

**Inferred lipid packing within and around prestin.** While detergent micelles have distinct properties from the lipid bilayer, we note an intimate relationship between the protein and micelle. Moreover, with the caveat that micelles may not be analogous to a bilayer, we note that the distance between the inner and outer "leaflets" of the micelle tends to vary across the protein's landscape (Fig. 5A, B). As indicated in Fig. 5, the micelle density shows conspicuous depressions around a region spanning TMs 6, 7, 12 (helices from gate domain). Measurements of the micelle's thickness indicates a locally thinned micelle in the vicinity of TMs 6-7. Thus, the distance between the micelle's "leaflets" is locally reduced to about 3.4 nm around the TM6 helix in comparison with a 4.4-nm thickness at the central part of the micelle. Supplementary Fig. 8 shows a corresponding variation in prestin's surface hydrophobicity across its transmembrane domain.

Compared with other members of the extended transporter family, the two protomers of the prestin dimer show minimal interactions between the transmembrane domains as is the case with Slc26a9. Thus, the space between the membrane domains of the two protomers is filled by the detergent micelle (Fig. 5A, B). In addition, we identified a number of amorphous densities (Supplementary Fig. 9A, B). These amorphous densities were analogous to similar densities found in Slc26a9 at 2.6 Å (see

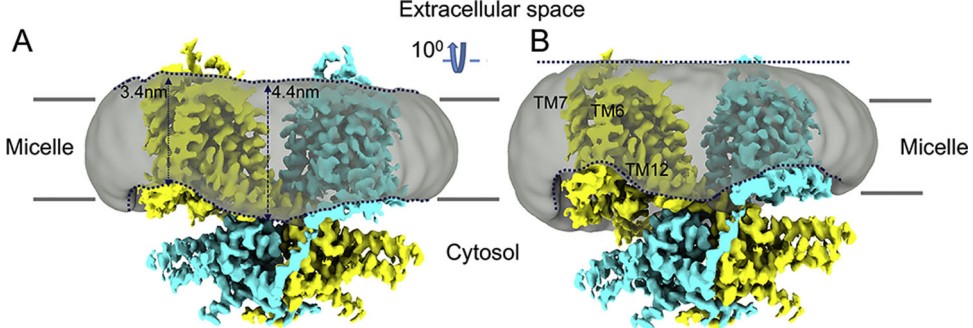

**Fig. 5 Interaction between prestin and the surrounding micelle. A** Cryo-EM map for the gerbil prestin structure (colored by subunit in cyan and yellow, respectively) showing the surrounding micelle as a gray transparent surface. The dashed line (in navy) follows the micelle's boundaries. The micelle belt around the transmembrane domain of prestin is locally distorted (thinned) at the cytosolic side close to TM6 and TM12 helices. The approximate thickness of the micelle in these regions is indicated by black arrows. The transmembrane helices 6, 7, and 12 are labeled. **B** As in (**A**) but tilted by ~10° to better illustrate the micelle's distortions at the cytosolic side. See also Supplementary Fig. 9.

Supplementary Fig. S3, b in[30]). We interpreted these densities as lipids.

These data raise a number of issues pertaining to lipid effects on the protein's function. For example, perhaps the variations in "leaflet" thickness are a reason for the shallow voltage dependence of prestin's NLC. We consider the influence of the surrounding lipid bilayer below.

## Discussion

We found that the overall molecular architecture of prestin resembles the previously reported cryo-EM structures of full-length human SLC26A9 (29) and of a shorter version of mouse Slc26a9 protein[29,30], whose functions are to facilitate the transport of anions. Slc26a9 allows entry of the substrates (anions) from the cytosol through the so-called "inward-open" conformation, which shows an opening between the "core" and the "gate" domains as described in the solved cryo-EM structures[29,30]. Intriguingly, we found that the membrane-spanning portion of prestin does not conform to the "inward-open" conformation of Slc26a9 (PDB ID, 7CH1). Rather, prestin has a more compact conformation, closer to the "intermediate" conformation of Slc26a9 (PDB ID, 6RTF), which shows the cytosol-exposed space between the "core" and the "gate" domains to be tighter. We reason that our prestin conformation solved by cryo-EM represents the contracted state of the protein at 0 mV.

Since submission of our paper, we note publication of a paper that presents the structure of human prestin in the presence of $Cl^-$ and $SO_4^{2-}$ anions[45]. Our structure in the presence of $Cl^-$ is highly concordant with that of Ge et al. (2021)[45] (both in detergent and nanodisc), which is reassuring. Thus, the RMSD values between our structure and the structure of human prestin in $Cl^-$ with detergent (PDB ID, 7LGU) and with nanodiscs (PDB ID, 7LGW) are nearly identical with Cα RMSDs of ~0.800 Å and ~0.838 over 1348 residues, respectively (Supplementary Fig. 11). The transmembrane helices and most of the loops connecting the TM helices remain at similar positions in gerbil prestin. The most prominent structural differences are at a region spanning residues (624–637) in the STAS domain, at the loop connecting TM7 and TM8 (residues 317–337), and especially at the cytosolic N-terminus (spanning residues 37–48), which shifts inside relative to the corresponding region in the human prestin structure (Supplementary Fig. 11B–E). These areas are the least well resolved in the density map of gerbil prestin, which suggests increased flexibility at these regions. Overall, prestin from gerbil in the presence of $Cl^-$ and detergent adopts a structural conformation very similar to the contracted conformation captured for human prestin.

Ge et al. also show prestin structures in the presence of $SO_4^{2-}$ (PDB ID 7LH2 RMSDs (Cα over 1348 residues) of 1.4 Å compared with our structure in $Cl^-$) that they ascribe as in the expanded state owing to the depolarizing shift in voltage dependence in the presence of this anion[45]. We would caution against such interpretation as the total charge movement ($Q_{max}$) in the presence of $SO_4^{2-}$ is reduced by ~2/3[24]. Since unitary charge is also reduced by an ~30% in $SO_4^{2-}$, these data would argue that 50% of motors are in an inactive or unresponsive state[24]. Moreover, the ~30% reduction in unitary charge movement would mean that there is incomplete movement of the protein molecules that do show voltage-dependent movement. In a similar vein, we find that the increase in linear capacitance in the presence of salicylate to be twice that of voltage-induced expansion[46,47], again making structures in its presence difficult to interpret in light of voltage-induced changes. These data together speak to the urgency in obtaining prestin's structure in changing voltage alone with physiological anions.

We have compared our prestin cryo-EM structure to the predicted structure by the AlphaFold algorithm and find that despite overall similar topology, the pairwise $C_{alpha}$ RMSD calculated over the number of residues resolved in the cryo-EM structure is 7.3 Å.

The essential work of the voltage-dependent protein prestin dwells at frequencies where mammals can hear, driven by OHC AC-receptor potentials. That is, the protein works in the kilohertz range of conformational change, with high-frequency measures of NLC reporting indirectly on those motions[4]. Cryo-EM structures of prestin, which necessarily define one or more steady-state conformations of the protein, may tell us little about its high-frequency physiology. Indeed, we cannot be sure that any identified state is actually occupied during high-frequency voltage stimulation, where molecular interactions, e.g., at the lipid–protein interface (see Fig. 5), may be influenced by rate (frequency) itself. The stretched-exponential nature of prestin's NLC likely reflects such interactions[48]. Nevertheless, static structures can inform on some key questions concerning prestin's molecular behavior.

Given the anion-binding pocket structural features (not assumed from other family members), we can assign relevance to prior and present mutational perturbations. Thus, we present data here that $Cl^-$ likely does not function as prestin's extrinsic voltage sensor, evidenced by the loss of anion sensitivity with structurally driven mutations in the anion-binding site. Instead, prestin possesses charged residues important for voltage sensing, akin to other voltage-sensing membrane proteins[19,49]. The distribution of twelve charged residues that sense voltage in prestin

has important implications. Seven of the 12 residues are located in the gate/dimerization domain in TMs 5, 6, and 12 that are modeled to move minimally in the elevator model[19,27,28]. Notably, all but two of the 12 residues lie in the intracellular halves of the TM domains or in the intracellular loops connecting these TM domains (Supplementary Fig. 10). In contrast, all but one of the five charged residues that have no effect on voltage sensing lie close to the extracellular halves of the TM domain or in the loops connecting these TM domains. This ineffectual group includes a charged residue in TM3 (R150) that is modeled to move significantly in the transporter cycle[19,27,28]. Together, these data suggest that contrary to functional expectations based solely on structural similarity between prestin and Slc26a9, electromechanical behavior in prestin is fundamentally different to transporter movements and is concentrated in proximity to the intracellular opening. It should be noted that majority of the charged residues responsible for NLC in prestin are also conserved in many of the other SLC26 transporter family members, including Slc26a6, that do not show NLC. In agreement with our conclusions, the recent paper by Ge et al.[45] noted that the anion binding residues moved 1–2 Å toward the cytoplasmic surface in the presence of $SO_4^{2-}$. Of course, these data would refute the extrinsic voltage-sensor hypothesis, as well, where movement of anions in the expanded state was predicted to be 25 Å in the opposite direction[45].

The wide dispersion of residue charges in several transmembrane domains that contribute to NLC and the inferred uneven lipid packing may underlie prestin's shallow voltage dependence. Indeed, the influence of lipids on prestin performance is well documented[50–53]. In agreement with the area motor model of prestin activity[54,55], we previously identified an augmentation of linear capacitance ($\Delta C_{sa}$) during hyperpolarization in prestin-transfected cells, our inducible prestin cell line (Fig. 3A) and OHCs that likely reflects an increase in membrane-surface area and bilayer thinning accompanying movement of prestin into the expanded state[46,47]. Thus, we conclude that the compact state that we observe structurally corresponds to minimal membrane-surface area and maximal membrane thickness. The converse would be expected for the expanded state. In this regard, although salicylate blocks NLC and eM, the effect it has on changes in linear capacitance ($\Delta C_{sa}$) indicates that it does not produce a natural state of prestin that is normally driven by voltage. That is, in the presence of salicylate, the area occupied by prestin, as indicated by $\Delta C_{sa}$, is doubled that produced in its absence when driven by voltage[46,47]. Thus, we might expect that direct voltage-driven conformational changes in prestin could differ from unphysiological (e.g., salicylate or $SO_4^{2-}$) anion-induced steady state cryo-EM structures. Interestingly, we have recently shown that the $\Delta C_{sa}$ component of membrane capacitance exists only in the real (capacitive) part of complex NLC, not in the imaginary (conductive) part, ostensibly linking it directly to membrane bilayer influence rather than prestin charge movement[56].

Interestingly, we note several amorphous densities in our 3.6-Å map, which correspond to identified lipids in human prestin (PDB ID, 7LGU). Some of these lipids in our structure are found between transmembrane helices (see Supplementary Fig. 9E–I). Since lipid pockets like these have been shown to influence protein–lipid interactions in the mechanosensitive Mscl channel[57], we reason that they may serve a similar function with prestin that shows similar sensitivity to membrane tension[11,58].

We have determined that prestin is dimeric as our previous studies have suggested[32,59], although earlier reports asserted that prestin functions as a tetramer[60,61]. In Slc26a9, three features were identified as important for dimerization and likely are pertinent for prestin considering the proteins' remarkable similarity. These include (1) interactions between individual

protomers exemplified by the valine zipper in TM14, (2) interactions between the C-terminal STAS domain of one protomer and the TM domain of the other and (3) an antiparallel beta-strand between the N terminus of both protomers. Prestin shows each of these same interactions with the valine zipper being replaced by leucine and isoleucine residues. The antiparallel beta-sheet between the N-terminal residues 15–20 of each protomer in prestin is likely critical for dimerization. Indeed, sequential deletion of residues 11–21 resulted in progressive loss of NLC, and loss of FRET signal confirming the loss of dimerization[32]. These data also argue that dimerization is critical for NLC, just as dimerization is important for transporter function in UraA[12,31].

The C terminus of prestin has high homology to the previously determined X-ray crystal structure of the C-terminal STAS domain (PDB ID, 3LLO) lacking the unstructured loop[35]. Thus, the many interpretations made by the Battistutta group are likely to therefore apply, including confirmation in our structure of the orientation of the STAS domain to the transmembrane domain, and the importance of the alpha-5 helix in stabilizing the core beta-sheets suggested by truncation experiments[32,62]. A significant difference was in the first alpha-helix that is parallel to the second alpha-helix as in bacterial ASA (anti-sigma factor antagonistic) proteins[63], and deviated by a 30° angle in the crystal structure[35]. This is likely due to the unstructured loop at the end of the first alpha-helix that was lacking in the crystal structure.

Finally, structural details of the C-terminal STAS domain can shed light on modulation of prestin by binding partners, for example, calcium/calmodulin[64]. Recently, calmodulin has been shown to bind to the STAS domain[65]. Such interactions may have physiological significance, since it was reported that $Ca^{2+}$/calmodulin shifts the operating voltage of prestin, namely $V_h$[66]. However, it was subsequently shown that shifts in $V_h$ were not due to direct action on prestin, but rather resulting from indirect tension effects on prestin due to OHC swelling[67]. The influence of the STAS domain on prestin's voltage-dependent activity remains an open question.

OHC electromotility was discovered in 1985[68,69], and 15 years later, prestin was identified as the protein responsible for the OHC's unique role in cochlear amplification[10,11,70]. During the intervening years, enormous detail into the protein's function has been obtained, see[26]. Recent homology modeling of prestin[25], based on presumed similarity to other family members, and confirmed in our structural data, has moved us closer to understanding prestin's electromechanical behavior. Indeed, the cryo-EM solution we provide here will permit us to rigorously interpret past studies and design experiments to more fully understand prestin's role in mammalian hearing. Given the remarkable similarity of prestin structure revealed in this present study to that of the inside-open/intermediate state of Slc26a9, key questions remain as to why the two proteins differ in their function. Indeed, established functional observations on prestin may have already identified key differences, for example, the observed negative cooperativity among the densely packed proteins interacting through membrane lipids (10,000/$\mu m^2$ in OHCs)[53,71], and subplasmalemmal cytoskeletal interactions with prestin[72]. Though key, prestin likely is a partner in the machinery that boosts our hearing abilities. Imperative in the overall effort to understand prestin is the need for alternative structures of prestin and other family members that are evoked by appropriate physiological stimuli, e.g., voltage in the case of prestin.

## Methods

**Prestin expression and purification**. The full-length prestin from gerbil (*Meriones unguiculatus*, Genbank accession number AF230376) was purified from a tetracycline-inducible stable HEK 293 cell line[37]. In establishing this cell line (16C), full-length gerbil prestin cDNA (a gift from J. Zheng and P. Dallos) tagged at its C

terminus with enhanced yellow fluorescent protein (EYFP) was inserted into the multiple cloning site of pcDNA4/TO/myc-HisC that allowed purification using Ni affinity.

Cells were grown in DMEM media supplemented with 1 mM L-glutamine, 100 U ml$^{-1}$ penicillin/streptomycin, 10% FBS, and 1 mM sodium pyruvate. About 4 µg/ml of blasticidin and 130 µg/ml of zeocin were supplemented in the growth media to maintain prestin expression. Cells were harvested 48 h after tetracycline (1 µg/ml) was added to the cell-growth medium to induce prestin expression.

Cell pellets from 20 T175 flasks were harvested by centrifugation at 1000 g for 10 min, washed with PBS, and then resuspended in 5 ml of resuspension buffer (25 mM HEPES, pH 7.4, 200 mM NaCl, 5% glycerol, 2 mM CaCl$_2$, 10 µg/ml$^{-1}$ DNase I, and 1 protease inhibitor) (complete EDTA-free, Roche) for each gram of pellet. About 2% (wt/vol final concentration) digitonin (Anatrace) powder was directly dissolved in the cell resuspension, and the mixture was incubated for 1.5 h under gentle agitation (rocking) at 4 °C. Insoluble material was removed by centrifugation at 160,000 g for 50 min (Beckman L90-XP ultracentrifuge with a 50.2 Ti rotor). The supernatant was passed through a 0.45 µm filter. In all, 10 mM imidazole (final concentration) and 1 ml Ni-NTA resin (Qiagen) prewashed in 25 mM Hepes, pH 7.4, 200 mM NaCl, 5% glycerol, 2mM CaCl$_2$, and 0.02% GDN (synthetic digitonin substitute glyco-diosgenin, Antrace) were added to the filtered supernatant and incubated with end-over-end rocking agitation at 4 °C for 2 h. The resin was collected using a bench Eppendorf microcentrifuge and washed sequentially with high-salt buffer Buffer A (25 mM HEPES, pH 7.4, 500 mM NaCl, 5% glycerol, and 0.02% GDN) and 25ml Buffer B (25 mM HEPES, pH 7.4, 200 mM NaCl, 10 mM imidazole, 5% glycerol, and 0.02% GDN). The protein was eluted in 1.5 ml Buffer C (25 mM HEPES, pH 7.4, 200 mM NaCl, 250 mM imidazole, 5% glycerol, and 0.02% GDN). The 1.5 ml eluted protein was concentrated to 500 µl, passed through a 0.22 µm filter, and loaded onto a FSEC column (Superdex 200 Increase 10/300 GL column, on a Shimadzu FPLC system) equilibrated with gel-filtration buffer (10 mM HEPES, 200 mM NaCl, and 0.02% GDN, pH 7.4). Two 0.5 ml fractions corresponding to the fluorescent (excitation 488 nm, emission 535 nm) peak and A280 peak were collected and concentrated using an Amicon Ultra centrifugal filter with a molecular weight cutoff 100 KDa and used for freezing grids.

Various combinations of detergents, including CHAPS, DM, DDM, and LMNG, were used to solubilize prestin. These detergents invariably resulted in broad peaks of the protein evidenced on the FSEC profile. We settled on the combination of digitonin/GDN that consistently gave us monodisperse profiles that were confirmed with uniform particles in negative-stain electron-microscopy images.

**Sample preparation and data acquisition.** An aliquot of four microliters of purified prestin (at a concentration of approximately 2 mg/ml) was applied to glow-discharged Quantifoil holey carbon grids (Gold R2/1, 200 mesh) overlaid with an additional 2-nm carbon layer (Electron Microscopy Sciences). The grids were blotted for 3–5 s and plunge-frozen in liquid ethane using a Vitrobot Mark IV (FEI) instrument with the chamber maintained at 10° C and 100% humidity.

Cryo-EM micrograph movies were recorded using the SerialEM software (v3.8) on a Titan Krios G2 transmission electron microscope (Thermo Fisher/FEI) operated at a voltage of 300 kV and equipped with a K3 Summit direct electron detector (Gatan, Pleasanton, CA). A quantum-energy filter with a 20-eV slit width (Gatan) was used to remove the inelastically scattered electrons. In total 4680 dose-fractionated super-resolution movies with 36 images per stack were recorded. The cryo-EM movies were recorded with a defocus varied from −1.15 to −2.15 µm at a nominal magnification of 81,000x (corresponding to 0.534 Å per physical pixel). The counting rate was 17.5 e$^-$/physical pix/s. The total exposure time was 3.6 s per exposure with a total dose of ~54 e$^-$/Å$^2$.

**Data processing.** Data processing was carried out with Relion 3.1[73], except as noted. Movie frames were gain-normalized and motion-corrected using Motion-Cor2 (v1.3.2)[74] with a binning factor of 2 and dividing micrographs into 4 × 4 patches. The dose-weighted, motion-corrected micrographs (the sum of 36 movie frames) were used for all image-processing steps, except for defocus determination. The contrast transfer function (CTF) calculation was performed with CTFFIND4.1 (as implemented in Relion3.1)[75] on movie sums that were motion-corrected but not dose-weighted. About 3000 particles were manually picked and subjected to 2D reference-free classification in Relion 3.1[73].

Classes showing good signal (representing ~1100 particles) were chosen as references for automated particle picking in Relion 3.1, yielding a dataset of ~1,377,109 particles. Several rounds of 2D and 3D classification (carried out without application of symmetry) were used to remove unsuitable particles, leaving 111,863 particles that were used for structural determination with imposed C2 symmetry in Relion 3.1. Bayesian polishing[73], followed by per-particle CTF refinement, 3D autorefinement, micelle-density subtraction and postprocessing generated a map that had an estimated resolution of ~3.6 Å according to the Fourier shell correlation (FSC) = 0.143 criterion.

**Model building.** The Slc26a9 cryo-EM intermediate structure (PDB ID: 6RTF) (which is a polyalanine trace) was rigid-body docked into the prestin cryo-EM map

and fitted using Chimera[76]. Next, the backbone was real-space-refined in Phenix (v 1.19.2)[77] and adjusted in COOT (v 0.8.9.1)[78] by manually going through the entire protomer chains. Sequence assignment was guided mainly by bulky residues such as Phe, Tyr, Trp, and Arg, and secondary structure predictions. Side chains in areas of the map with insufficient density were left as alanine. The model was refined through several rounds of model building in COOT and real-space refinement in PHENIX[77] with secondary structure and geometry restraints. The model was validated using MolProbity[79]. Details on the statistics of cryo-EM data collection and structure determination are presented in Supplementary Table 1. Figures were prepared using Chimera (v 1.12)[76] and ChimeraX (v 0.9)[80].

**Electrophysiological recording.** Recordings were made of transiently transfected CHO (or HEK cells 48 h after tetracycline induction) using a whole-cell configuration at room temperature using an Axon 200B amplifier (Molecular Devices, Sunnyvale, CA), as described previously. Cells were recorded 48–72 h after tetracycline induction or transfection (Fugene (Promega) according to the manufacturer's instructions) to allow for stable measurement of current and NLC. Mutations were introduced in gerbil prestin YFP using the QuickChange Mutagenesis Kit (Agilent) according to the manufacturer's instructions. We recorded data from control cells on each day of experimentation, and group those within the same timeframes. Both controls were prestin–YFP. Experiments with S398E were done first. Experiments with the second set of mutants were done afterward, about 3 weeks after S398E. Since the experiments were done separately, they are presented as such. In specific instances, blinding was not possible since the experimenter was responsible for both transfection of the cells and recording from them. Where the experimenters were different, electrophysiological experimenters were blinded to the identity of the constructs transfected into cells being recorded. Blinding is effective to avoid artificial group difference that is caused by performance bias. We reported mutants that show similar z. Unblinding has no bias on null results.

Randomization was not done since electrophysiological recordings were performed in batches of cells transfected with the same construct. The benefit of randomization is to eliminate the bias from known and unknown confounding variables that distribute unbalanced between comparison groups. We compared NLC parameters on cultured cells rather than human or animal subjects. Randomization is not applicable to our study. The number of cells is standard in the field. For those results with $p > 0.05$ in our paper, the absolute differences are very small and in the normal range of NLC parameters. Thus, it is statistically insignificant. Increasing sample size may improve the SE but will not change the absolute difference and alter our conclusion. All recordings with Rm > 300 mOhms and Rs < 10 mOhms were included in the analysis. The standard bath-solution components were (in mM) 100 NaCl (TEA)–Cl 20, CsCl 20, CoCl$_2$ 2, MgCl$_2$ 2, and Hepes 5, pH 7.2. In addition, 20 mM NaSCN was substituted for current recordings with cells transiently transfected with Slc26a9. The pipette solution contained (in mM): NaCl 100, CsCl 20, EGTA 5, MgCl$_2$ 2, and Hepes 10, pH 7.2. Osmolarity was adjusted to 300 ± 2 mOsm with dextrose. After whole-cell configuration was achieved in extracellular NaCl a ramp protocol recorded to confirm baseline NLC and currents. Pipettes had resistances of 3–5 MΩ. Gigaohm seals were made and stray capacitance was balanced out with amplifier circuitry prior to establishing whole-cell conditions. A Nikon Eclipse E600-FN microscope with 40× water-immersion lens was used to observe cells during voltage clamp. Data were low-pass filtered at 10 kHz and digitized at 100 kHz with a Digidata 1320A.

Command delivery and data collections were carried out with a Windows-based whole-cell voltage-clamp program, jClamp (v 31.7.0, Scisoft, East Haven, CT), using a Digidata 1322A (Axon Instruments). A continuous high-resolution 2-sine voltage command was used, cell capacitance and current being extracted synchronously. In order to extract Boltzmann parameters, capacitance-voltage data were fit to the first derivative of a two-state Boltzmann function.

$$C_m = NLC + C_{sa} + C_{lin} = Q_{max} \frac{ze}{k_B T} \frac{b}{(1+b)^2} + C_{sa} + C_{lin} \quad (1)$$

$$\text{where } b = \exp\left(-ze \frac{V_m - V_h}{k_B T}\right), C_{sa} = \frac{\Delta C_{sa}}{(1+b^{-1})} \quad (2)$$

$Q_{max}$ is the maximum nonlinear charge moved, $V_h$ is voltage at peak capacitance or equivalently, at half-maximum charge transfer, $V_m$ is membrane potential, $z$ is valence, $C_{lin}$ is linear membrane capacitance, e is electron charge, $k_B$ is Boltzmann's constant, and T is absolute temperature. $C_{sa}$ is a component of capacitance that characterizes sigmoidal changes in specific membrane capacitance, with $\Delta C_{sa}$ referring to the maximal change at very negative voltages[46,47]. For fits of NLC in transiently transfected cells, $C_{sa}$ was not included. $Q_{sp}$ the specific charge density, is the total charge moved ($Q_{max}$) normalized to linear capacitance. Voltages were corrected for series-resistance voltage drop. Separately, in specific experiments gating currents were also determined using voltage steps (50-ms duration) from −100 mV to 150 mV, with 20 mV step increments. Where anions were substituted, local perfusion of the cells was estimated to give rise to small junctional potentials (JPCalc function in pClamp). Since these numbers were small, no corrections were made to the IV plots. For solution-perfusion experiments, we used a QMM perfusion system (ALA Scientific, Instruments, Westbury, NY). The manifold's output tip was 200 µm placed 1 mm from the cell, and the flow rate increased by an

applied pressure of approximately 20 kPa. Statistical analysis was done with SAS software (SAS Institute Inc, NC).

**Reporting summary**. Further information on research design is available in the Nature Research Reporting Summary linked to this article.

## Data availability
Source data are provided with this paper. Cryo-EM map and atomic coordinates have been deposited to the EMDB and PDB with accession codes: EMD-25442 and PDB ID 7SUN. Source data are provided with this paper.

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

## Acknowledgements

This research was supported by NIH-NIDCD R01 DC016318 (JSS) and R01 DC008130 (JSS, DN). The authors wish to thank Dr. Fred Sigworth for advice and helpful comments. This work used the electron microscopy facilities from Yale School of Medicine. We would like to thank Dr. Shenping Wu, Dr. Marc Llaguno, Dr. Xinran Liu, Kaifeng Zhou, and Dr. Jianfeng Lin for access to the TEM infrastructure and for managing Yale's electron microscopes.

## Author contributions

C.B. designed experiments, optimized the protein-purification protocol, and carried out the cryo-EM experiments: grid freezing, single particle-data collection, data processing, and structure determination, and wrote the paper. Q.S. designed experiments, expressed and purified proteins, and performed mutagenesis. J.P. and W.T. designed experiments and performed electrophysiological experiments and analysis. D.S.N. and J.S.S. designed experiments, analyzed data, and wrote the paper.

## Competing interests

The authors declare no competing interests.
