## [Peer Review File · Nature Communications]

Single-particle cryo-EM structure of the outer hair cell motor protein prestin.REVIEWER COMMENTS

Reviewer #1 (Remarks to the Author):

The authors of this study report the structure of the outer hair cell motor protein prestin (Slc26a5) from gerbil determined by cryo-EM at 3.9 Å. Complementary functional experiments demonstrate that the protein, unlike its close characterized homologue SLC26a9, shows the typical non-linear capacitance (NLC), which is a hallmark of this unusual member of a family of anion transport proteins. The study has been carried out by the group of a long-term expert in the field and is generally of high quality.

Although the manuscript is one of three studies on the subject that have recently been published in peer reviewed journals and on preprint servers, all conclusions are sound and were derived independently. While at somewhat lower resolution than its competing studies, the quality of the structure is sufficiently high to support the claims made in the publication. I thus think that the manuscript is strong and a suitable candidate for publication in Nature Communications after some revisions.

Suggested changes:

In the discussion, the authors compare their structure to the prestin model obtained from the alpha-fold database. Although interesting, it might be more informative if the comparison would be made to the human prestin structures obtained by Ge et al. (doi: 10.1016/j.cell.2021.07.034), which have already been released in the PDB.

Along the same line it would be interesting if the authors would briefly relate their conclusions to the conclusions drawn in the manuscript by Ge et al. which was recently published in Cell and, if they wish, also to the study of Bavi et al, which is available at bioRxiv.

On page 5 the authors conclude that a bound Cl⁻ would not contribute to the voltage sensing of prestin based on an experiment that replaces a serine (S398), that is known to interact with the inhibitor salicylate, to a glutamate, which abolishes the inhibition by salicylate but still confers electromotility.

My two questions are: 1. Is it known whether this mutation does also inhibit chloride binding?

2. Would not the introduction of a negative charge at the binding site (by the glutamate sidechain) rather emphasize the importance of an anion in this region to confer electromotility?

I thus wonder on which basis the authors exclude the possibility that a bound Cl⁻ would contribute to a delocalized voltage sensor (together with other residues in the transmembrane part of the protein). Would not the bound negatively charged Cl⁻ necessarily contribute to the distribution of conformations in response to voltage as soon as it is located within the transmembrane electric field (which would likely be the case upon any conformational change of the transmembrane domain)?

Minor:

Introduction:

Page 1 line 3:

Although the SLC26, SLC4 and SLC23 protein families share a similar fold of their membrane inserted domain, I do not think that this qualifies them as part of the same protein family, as they lack any obvious sequence relationship. Prestin is thus a member of the SLC26 family.

Page 1 line 14:

There is no evidence that any of the described family members would work as ion channels. Instead, the presumed channels of the SLC26 family are probably functioning as uncoupled transporters.

Page 1 line 22:

To my knowledge there is no structure-based proposal that SLC26 transporters would work by a rocker-switch mechanism, which is characteristic for transporters of the major facilitator super-family.

Page 5, bottom:

The authors claim that the largest part of the dimer interface would be constituted by contacts between the STAS domain of one and the TM domain of the interacting subunit. To this end it would be interesting to know which fraction of the surface is buried between the two STAS domains in comparison to STAS-TM and STAS N-terminus interactions.

Page 8 line 5,

Although it is reasonable to assume that, under the conditions applied during purification, prestin would be in a contracted state in a cellular environment, the equilibrium between states can be altered upon solubilization in detergents.

Reviewer #2 (Remarks to the Author):

In this manuscript Butan et al. present one cryo-EM structure of gerbil prestin and discuss its significance also on the basis of new electrophysiological data. Unfortunately, very recently, a paper presenting four different cryo-EM structures of human prestin has been published in Cell (Ge et al., Molecular mechanism of prestin electromotive signal amplification, 2021, Cell 184, 4669–4679).

Gerbil prestin (Butan et al.) was expressed in HEK293 cell line and purified in glycol-diogenin (GDN) micelles. A similar strategy was used for human prestin (Ge et al.) for which it was also verified that the structure is essentially the same in lipid nanodiscs. In Butan et al., functional implications are based on

one structure at 4 Å resolution while in Ge et al. on four different structures (with bound Cl⁻ in GDN micelles, with bound Cl⁻ in lipid nanodiscs, with bound sulfate and with bound salicylate), with the best structure at 2.3 Å resolution.

As far as I understand from the reading of the manuscript the structure of gerbil prestin presented by Butan et al. does not present significant new features compared with that presented by Ge et al., that is, it does not carry new original pieces of information. Instead, the lower resolution of the gerbil structure hampers to catch interesting features described for human prestin as the chloride anion bound in the “active site” or the cholesterol and lipid molecules surrounding prestin. Butan et al. infer that the obtained structure is in the contracted state. Ge et al., taking advantage from the analysis of the four different structures, were able to characterize a contracted and an expanded state for prestin, revealing also the extensive prestin-membrane interactions and providing a glimpse of how prestin inserts into the membrane, couples to deformation of the bulk membrane, and induces membrane curvature (on the basis of over a hundred well-resolved lipid molecules, including two cholesterol molecules in the space between the TMDs).

Both Butan et al. and Ge et al. came to a similar conclusion, that is their data are more consistent with an intrinsic voltage sensor model for prestin, where partial charges of multiple sidechains distributed within the motor protein move by smaller amounts.

In conclusion, in the current version of the manuscript, structural results (beyond their soundness) and functional inferences presented by Butan et al. lack the necessary novelty and are less informative compared with those already published in the paper by Ge et al., (lower resolution, one structure vs four different structures etc.), which were able to get better and more robust insights into the structural basis of prestin function related to OHC electromotility.

Other comments:

- In the Materials and Methods section “Prestin Expression and Purification”, it seems that authors left the EYFP tag linked to prestin and used this un-cleaved construct for freezing grids. Indeed, in Figure 1 Supplement 1, EYFP-Prestin is clearly not cleaved. Authors should comment this point. Additionally, the procedure of sample preparation is not explained in details, for instance they should comment the choice of the GDN micelles.

- In supplementary Figure 1 Supplement 2 authors could include also the Euler angle distribution of particles for the cryo-EM reconstruction to highlight distinct preferential orientations.

Reviewer #3 (Remarks to the Author):

For context, this paper comes as part of a surge of 3 papers, reporting the eagerly awaited cryo-EM structure of the OHC motor protein prestin (SLC26A5).

Here, Butan et al. report a single structure of prestin from the gerbil.

Taken together, the structure (of reasonable, but not highest resolution) shows the basic features that have been expected, based on recent experimental structures of the homolog, SLC26A9, and a previous modeling effort with prestin. These features confirmed by the new structure include the 7TMIR fold, global domain structure (core/gate), identity of the anion binding site and dimerization involving the domain-swapped STAS arrangement.

From there, the authors aim at understanding the mechanism by which prestin generates cellular motility ('piezoelectric behavior'). Although the manuscript lacks clarity in this respect, it appears that the tacit assumption is that electromotility results from transition between (2 or likely more) conformations that differ mechanically. Given that only a single state is described here (unlike in the 'companion' papers), no straightforward conclusions can be made as to the mechanism.

However, the authors argue that they can exclude a transporter-like mechanism (for SLC26 transporters most likely an elevator-like movement of the core-domain) for electromotility. This is the most questionable aspect of the manuscript, presented in a difficult-to-follow manner, and conclusions rest on rather circumstantial evidence.

First, it is claimed that the electrical partial charges that contribute to the voltage sensitivity of electromotility are distributed across domains that are not expected to move in the elevator movement that has been postulated for SLC26 transporters. However, the actual transport dynamics have not been defined experimentally for any SLC26 transporter yet, hence the absence of movement in any domain is speculative.

Further, the conclusions rely mostly on previous findings from this group, where neutralizing 12 distinct residues individually (by mutation) was reported to reduce voltage dependence. Those effects should be interpreted very conservatively, as the overall sensor charge is well below 1 e and hence the contribution of each charged residue must be tiny, which is also true for the expected effects on NLC voltage dependence. Conspicuously, the sensor charges of the binding site mutations reported in the present manuscript all fall into the range considered to indicate reduced sensor charge in the previous paper - but here this is claimed to indicate 'normal unitary gating charge'. Also, charged residues may also impact on apparent voltage dependence by sculpting the local electrical field experienced by the mobile 'voltage sensor' charges.

Also, I do not see how the presented binding site mutations exclude a contribution of the bound substrate anion to voltage sensor charge. It appears that the introduction of a glutamate into this site functionally replaces the extrinsic anion (but cannot be pushed out by salicylate) and the charge could move with the core domain in an elevator-like movement. Other mutations in the binding site are made, but how these experiments constrain the mechanism is unclear.

It is then shown that the homolog, SLC26A9, lacks charge movement and hence electromotility (see concern below). The rationale for presenting these experiments is unclear. In general, it should be

considered that electrogenicity and substrate charge vary across the SLC26 family, so the charge moved with the transport steps may differ despite a common transport mechanism expected from structural conservation. Specifically, SLC26A9 behaves like a chloride channel, and therefore may actually lack the transport transition. It may therefore not be the appropriate model for asking whether transport involves electromotility-like charge movement/NLC. On the contrary, the structurally most closely related SLC26 transporter (A5 from non-mammals) was reported to feature charge movement.

Other points

1. In the experiments on SLC26A9 charge movement, the expression levels are questionable. According to other papers (e.g. Walter et al., 2019), chloride currents of the full-length protein expressed transiently can be on the order of nA. Here, no current at all is reported and the authors have to use SCN as an alternative substrate to see any functional expression at all. Maybe the expression is simply too low to allow detection of NLC.

2. It is argued that the state resolved here represents the contracted one. This seems plausible, considering prestin's voltage dependence. However, given the lack of any membrane-like environment together with the biophysically well documented impact of membrane composition and tension on prestin's state distribution, this conclusion appears less certain. The lack of membrane environment might force the protein into any native (or even non-native) state.

Responses (in red) to Reviewer comments (in black)

First of all, we would like to thank the reviewers for making this a better paper.

Reviewer #1 (Remarks to the Author):

The authors of this study report the structure of the outer hair cell motor protein prestin (Slc26a5) from gerbil determined by cryo-EM at 3.9 Å. Complementary functional experiments demonstrate that the protein, unlike its close characterized homologue SLC26a9, shows the typical non-linear capacitance (NLC), which is a hallmark of this unusual member of a family of anion transport proteins. The study has been carried out by the group of a long-term expert in the field and is generally of high quality. Although the manuscript is one of three studies on the subject that have recently been published in peer reviewed journals and on preprint servers, all conclusions are sound and were derived independently. While at somewhat lower resolution than its competing studies, the quality of the structure is sufficiently high to support the claims made in the publication. I thus think that the manuscript is strong and a suitable candidate for publication in Nature Communications after some revisions.

We agree. Thank you.

Suggested changes:

In the discussion, the authors compare their structure to the prestin model obtained from the alpha-fold database. Although interesting, it might be more informative if the comparison would be made to the human prestin structures obtained by Ge et al. (doi: 10.1016/j.cell.2021.07.034), which have already been released in the PDB.

We agree since the Cell paper has since been published and the PDB made available (it was not available when we submitted). We wanted to see how well AlphaFold would do, since its release just prior to our submission has been an “earthquake” in the structural biology world.

We have now reported in the manuscript text the RMSD values for the structural alignment of the whole prestin structure with the structures reported by Ge et al. The structures of gerbil and human prestin align with a RMSD (Ca. over 1,348 residues) of 0.800 Å (PDB:7LGU, human prestin in NaCl) and of 1.4 Å (PDB:7LH2, human prestin in SO₄²⁻ and salicylate). This information has been included in the results and discussions.

Also, we have now included figures showing the superpositions of the gerbil prestin and the human prestin structure (PDB:7LGU) determined by Ge et al. (Supplementary Figure 7).

Along the same line it would be interesting if the authors would briefly relate their conclusions to the conclusions drawn in the manuscript by Ge et al. which was recently published in Cell and, if they wish, also to the study of Bavi et al, which is available at bioRxiv.

We agree since the Cell paper has since been published and the PDB made available. In brief, the manuscript by Ge et al., largely complements our own- The most prominent structural differences are at a

region spanning residues (624-637) in the STAS domain, at the loop connecting TM7 and TM8 (residues 317-337) and especially at the cytosolic N-terminus (spanning residues 37-48) which shifts inside the corresponding region in the human prestin structure. These areas are the least well resolved in the density map of gerbil prestin, which suggests increased flexibility at these regions. The 14 transmembrane helices in the gerbil structure remain at similar positions as the 14 transmembrane helices of the human prestin structure resolved in the contracted state. Based on these, we can conclude that our structure has captured a conformation similar to the structure of the contracted (PDB:7LGU, human prestin in NaCl) state reported by Ge et al.

We are not sure about the Bavi et al. paper in bioRxiv since it is not finalized, and we don't want to make claims that may not be valid once the paper is published. Moreover, the PDBs have not been made available to us.

We do not agree with Ge et al., that the structures in salicylate or SO_4^{2-} represent the voltage driven fully expanded state. Previous experiments have shown that the primary physiological driver of expansion and contraction is voltage; an effect that is modified by physiological concentrations of chloride.

*The differences in linear capacitance at hyperpolarized levels (increased linear capacitance indicating an expanded state) in the presence of salicylate is two-fold or greater than that with hyperpolarizing voltage alone (Santos-Sacchi and Navarrete, 2002; Santos-Sacchi and Song, 2014). Additionally, the total charge movement (NLC) in the presence of SO_4^{2-} is reduced by $\sim 2/3$ and taken together with a $1/3$ reduction in unitary gating charge in the presence of SO_4^{2-} (Rybalchenko and Santos-Sacchi, 2008), would mean that 50% of the molecular motors are inactive or unresponsive in the presence of SO_4^{2-} . To conclude, therefore, that the protein is in a normal expanded state, as is the case with voltage, is a **significant overreach**.*

Furthermore, since there is a $1/3$ reduction in unitary charge movement, it would suggest that the protein movements of the functional motors in presence of sulfate are significantly less than in the presence of chloride. The intriguing data here is that the anion binding site moves towards the intracellular surface with salicylate/ SO_4^{2-} . This is their coup de grâce against the extrinsic hypothesis. Of course, these data are in agreement with our decades long effort showing that chloride affects prestin through an allosteric-like mechanism and that it does not function as the extrinsic voltage sensor (one that Dominik Oliver, who initially proposed the extrinsic hypothesis, now agrees with (Gorbunov et al., 2018). It is important that the readers be aware of the literature that must dictate evaluations of the structure. We have previously commented on this necessity in an annotation posted on the eLife Slc26a9 structural paper of Dutzler's group (see https://hyp.is/sDYyJOV-Eemh6i_fc6x0tw/elifesciences.org/articles/46986).

On page 5 the authors conclude that a bound Cl^- would not contribute to the voltage sensing of prestin based on an experiment that replaces a serine (S398), that is known to interact with the inhibitor salicylate, to a glutamate, which abolishes the inhibition by salicylate but still confers electromotility. My two questions are: 1. Is it known whether this mutation does also inhibit chloride binding?

We can confidently assume that it would, since salicylate binds competitively with chloride (Oliver et al., 2001; Santos-Sacchi et al., 2006) and the mutant lacks salicylate sensitivity.

2. Would not the introduction of a negative charge at the binding site (by the glutamate sidechain) rather emphasize the importance of an anion in this region to confer electromotility?

We agree that it would, and we have always claimed that chloride binding promotes voltage dependency through an allosteric-like mechanism.

I thus wonder on which basis the authors exclude the possibility that a bound Cl⁻ would contribute to a delocalized voltage sensor (together with other residues in the transmembrane part of the protein). Would not the bound negatively charged Cl⁻ necessarily contribute to the distribution of conformations in response to voltage as soon as it is located within the transmembrane electric field (which would likely be the case upon any conformational change of the transmembrane domain)?

*We agree that the existence of chloride could theoretically alter the landscape. But the mutations that we have made indicate that expected anion binding alterations have no influence. Furthermore, the mutated residue charge is not expected to mirror chloride's location and displacement within the membrane field. In other words, the mutation charge movement would be expected to be different than the normal condition, but it is not. Moreover, mutation of additional residues (R399, Q94, Q97 and P136) that coordinate binding to chloride that were based on Slc26a9, and since confirmed in prestin in the Ge et al 2021 paper, have no effect on unitary charge movement. This is concordant with other data from the Ge et al paper, where they find a small movement of the chloride binding residues (1-2 Å) **towards** the intracellular surface (and not 25 Å **away** from the intracellular surface, as would be expected by the extrinsic sensor hypothesis) in the SO₄²⁻ bound state that they interpret as the expanded state.*

Minor:

Introduction:

Page 1 line 3:

Although the SLC26, SLC4 and SLC23 protein families share a similar fold of their membrane inserted domain, I do not think that this qualifies them as part of the same protein family, as they lack any obvious sequence relationship. Prestin is thus a member of the SLC26 family.

We agree. We have changed the text.

"... belongs to a diverse family of transporters that includes 9 members (Chang and Geertsma, 2017)."

Page 1 line 14:

There is no evidence that any of the described family members would work as ion channels. Instead, the presumed channels of the SLC26 family are probably functioning as uncoupled transporters.

This is our view too, since the mechanisms of ion channels and transporters significantly differ. However, comments from the other reviewers and published work in the literature indicate that this is a widely held view.

We have therefore changed the text as follows:

"...variably as coupled transporters and uncoupled transporters/ ion channels with..." .

Page 1 line 22:

To my knowledge there is no structure-based proposal that SLC26 transporters would work by a rocker-switch mechanism, which is characteristic for transporters of the major facilitator super-family.

We agree. The text has been modified.

“...With more structural information there have been competing visions of transporter mechanisms (elevator vs. rocker) in the related proteins that share similar structural folds (Drew and Boudker, 2016; Fici et al., 2017), although..”

Page 5, bottom:

The authors claim that the largest part of the dimer interface would be constituted by contacts between the STAS domain of one and the TM domain of the interacting subunit. To this end it would be interesting to know which fraction of the surface is buried between the two STAS domains in comparison to STAS -TM and STAS N-terminus interactions.

We agree with the referee, and we provide calculations of the buried surface areas between the different interfaces of the gerbil structure. The ranking is as it follows: the combined buried surface area between one STAS subunit (505-726) and the N-terminus (residues 13-75) of the opposing subunit is 1438 Å² (representing 19% of contacts), the combined buried surface area between one STAS subunit (505-726) and the TM-domain of the opposing subunit (residues 76-504) is 1293 Å² (17%), and the buried surface area between two STAS subunits (residues 505-726) is 1192 Å² (15 %). This information has been included in the first paragraph which describes the structure of gerbil prestin as well as in Supplemental Table 2 in the supplementary materials.

Page 8 line 5:

Although it is reasonable to assume that, under the conditions applied during purification, prestin would be in a contracted state in a cellular environment, the equilibrium between states can be altered upon solubilization in detergents.

This may be possible. However, we believe that isolated detergents cannot deliver tension to the structure similar to that we are able to provide in electrophysiological experiments. Thus, the near identical structures that we and Ge et al., (2021) find are most likely in the contracted state.

Reviewer #2 (Remarks to the Author):

In this manuscript Butan et al. present one cryo-EM structure of gerbil prestin and discuss its significance also on the basis of new electrophysiological data. Unfortunately, very recently, a paper presenting four different cryo-EM structures of human prestin has been published in Cell (Ge et al., Molecular mechanism of prestin electromotive signal amplification, 2021, Cell 184, 4669–4679).

Gerbil prestin (Butan et al.) was expressed in HEK293 cell line and purified in glycol-diogenin (GDN) micelles. A similar strategy was used for human prestin (Ge et al.) for which it was also verified that the structure is essentially the same in lipid nanodiscs. In Butan et al., functional implications are based on one structure at 4 Å resolution while in Ge et al. on four different structures (with bound Cl⁻ in GDN micelles, with bound Cl⁻ in lipid nanodiscs, with bound sulfate and with bound salicylate), with the best structure at 2.3 Å resolution.

As far as I understand from the reading of the manuscript the structure of gerbil prestin presented by Butan et al. does not present significant new features compared with that presented by Ge et al., that is, it does not carry new original pieces of information. Instead, the lower resolution of the gerbil structure hampers to catch interesting features described for human prestin as the chloride anion bound in the “active site” or the cholesterol and lipid molecules surrounding prestin. Butan et al. infer that the obtained structure is in the contracted state. Ge et al., taking advantage from the analysis of the

four different structures, were able to characterize a contracted and an expanded state for prestin, revealing also the extensive prestin-membrane interactions and providing a glimpse of how prestin inserts into the membrane, couples to deformation of the bulk membrane, and induces membrane curvature (on the basis of over a hundred well-resolved lipid molecules, including two cholesterol molecules in the space between the TMDs).

Both Butan et al. and Ge et al. came to a similar conclusion, that is their data are more consistent with an intrinsic voltage sensor model for prestin, where partial charges of multiple sidechains distributed within the motor protein move by smaller amounts.

In conclusion, in the current version of the manuscript, structural results (beyond their soundness) and functional inferences presented by Butan et al. lack the necessary novelty and are less informative compared with those already published in the paper by Ge et al., (lower resolution, one structure vs four different structures etc.), which were able to get better and more robust insights into the structural basis of prestin function related to OHC electromotility.

It must be understood that we did our work simultaneously and independently of Ge et al., (2021). Yes, the reviewer is right that the structures of human and gerbil are basically the same, with minor variations, which is reassuring. It is our interpretations buttressed by additional experiments that differ. Key points we have established that are not present in the Ge et al paper include the following:

- 1. Location of the voltage sensor.*
- 2. That transporter like movements are unlikely to be causative of prestin's electromotility based on absent gating charge movement in Slc26a9.*
- 3. The uneven distribution of lipid / detergent within the bilayer and the resultant uneven voltage across the protein that could help explain the shallow voltage gradient.*
- 4. The presence of membrane pockets between transmembrane helices that likely mediate protein lipid interactions much like the mechanosensitive channel MscL. The Ge et al paper suggest similarity to the membrane bending in Piezo. We too see similar small discrepancies in the inner and outer detergent leaflets. We do not think these discrepancies compare with that seen in Piezo, where the membrane bends by over 15°-20° largely due to the extra-membranous cap.*
- 5. The considerably narrower inner vestibule (evidenced by the reduced angle between TMs 13, 14 and TM 8) that increases the kinetic barrier to the passage of ions and could explain the greater currents with SCN⁻ that is easier to dehydrate and is smaller than the hydrated Cl⁻ ion in prestin. This is an important point since the narrow vestibule could explain the reduced transporter activity in prestin. It is the reduced transporter activity that then gave rise to the extrinsic hypothesis. We now provide a plausible alternative explanation for the reduced transporter activity, and do not have to evoke arrested hemitransporter movements – the basis of the extrinsic hypothesis.*

So we respectfully, though strongly, disagree with the reviewer's negative views.

During the review process, we have improved the resolution of the cryo-EM map of gerbil prestin by identifying a subset of more homogeneous particles from which we generated a structure with an overall resolution of 3.6 Å (according to the FSC=0.143 criterion). We have reported this improvement in the Materials and Methods section as well as in the Table S1 (Cryo-EM data collection, refinement and validation statistics).

The improved clarity of the 3.6 Å density map yielded a more accurate description of the prestin structure. There are no substantial differences between the 3.6 Å-derived model compared to the model derived from the previous ~4 Å, and does not affect our structural interpretations. The conclusions based on the

comparisons of 4 Å prestin structure to the two previously reported conformations of the SLC26A9 structure (PDBs: 7CH1 and 6RTF) remain valid.

We have also redone some of the figures in the manuscript to present the higher resolution density map as well as the re-refined model. The outline of the figures remains unchanged; the panels included into the new figures (describing the improved density map and the corresponding model) are essentially the same as those reported previously for our 4 Å prestin structure.

As explained in the answers to Referee #1 we have compared our structure to the structures reported by Ge et al, (2021). While structurally superpositioning the contracted state of human prestin (PDB:7LGU) with our structure we noticed that some of the lipids present in the reported human prestin structure (PDB:7LGU) fit well into some “unaccounted for” densities present in our prestin density map. The positions of these “unaccounted for” densities relative to the gerbil prestin densities are illustrated in the newly added Figure 5, Supplementary figure 2.

A similar observation was made by Chi et al., in their 2.6 Å structure of Slc26a9 (Cell discovery, 2020, Supplementary Fig. S3, panel b). They also observed several patches of rod-like non-protein densities. They commented on these non-protein densities observed in their 2.6 Å electron density-map: “Lipid or detergent molecules are observed between the TM domains of the two protomers. The lack of shape features makes it difficult to conclude whether this density is a phospholipid or a GDN molecule”.

Butan et al. infer that the obtained structure is in the contracted state. Ge et al., taking advantage from the analysis of the four different structures, were able to characterize a contracted and an expanded state for prestin, revealing also the extensive prestin-membrane interactions and providing a glimpse of how prestin inserts into the membrane, couples to deformation of the bulk membrane, and induces membrane curvature (on the basis of over a hundred well-resolved lipid molecules, including two cholesterol molecules in the space between the TMDs).

The structure in the Ge et al. paper in Cl⁻ and ours in Cl⁻ are the same and we both conclude they are in the contracted state based on the voltage sensitivity in chloride. Yes, their paper is nice and careful. However, we do not agree that all their states are normal voltage driven states, i.e., we do not agree that the expanded state observed with sulfate or salicylate is the same as that normally achieved with voltage perturbation, which is the **physiological** driver of conformational change in prestin.

We have responded to **Reviewer 1**, who invited comment on the comparison of ours and their structures and for our interpretations of the paper by Ge et al. We reiterate the previous response below.

The differences in linear capacitance at hyperpolarized levels (increased linear capacitance indicating an expanded state) in the presence of salicylate is two-fold or greater than that with hyperpolarizing voltage alone (Santos-Sacchi and Navarrete, 2002; Santos-Sacchi and Song, 2014). Additionally, the total charge movement (NLC) in the presence of SO₄²⁻ is reduced by ~2/3 and taken together with a 1/3 reduction in unitary gating charge in the presence of SO₄²⁻ (Rybalchenko and Santos-Sacchi, 2008), this would mean that 50% of the molecular motors are inactive or unresponsive in the presence of SO₄²⁻. To conclude, therefore, that the protein is in a normal expanded state, as is the case with voltage, is a **significant overreach**.

Furthermore, since there is a 1/3 reduction in unitary charge movement, it would suggest that the protein movements of the functional motors in presence of sulfate are significantly less than in the presence of chloride. The intriguing data here is that the anion binding site moves towards the intracellular surface with salicylate/SO₄²⁻. This is their coup de grâce against the extrinsic hypothesis. Of course, these data are in agreement with our decades long effort showing that chloride affects prestin through an allosteric-like mechanism and that it does not function as the extrinsic voltage sensor (one that Dominik Oliver, who initially proposed the extrinsic hypothesis, now agrees with (Gorbunov et al., 2018). It is important that the readers be aware of the literature that must dictate evaluations of the structure. We have previously commented on this necessity in an annotation posted on the eLife Slc26a9 structural paper of Dutlzer's group (see https://hyp.is/sDYyJOV-Eemh6i_fc6x0tw/elifesciences.org/articles/46986).

Now that we have had access to the Ge et al. structural information, it is not clear that they provide a "glimpse of how prestin inserts into the membrane" any different from what we observe, since their nanodisc and micelle structures, and ours in micelles, are essentially the same. The RMSD values of our structure and theirs (C α over 1,348 residues) in detergent is 0.800 Å (PDB: 7LGU), and our structure and theirs in nanodiscs is 0.838 Å (PDB: 7LGW). This may result because native OHC membrane lipids are not evaluated.

Other comments:

- In the Materials and Methods section "Prestin Expression and Purification", it seems that authors left the EYFP tag linked to prestin and used this un-cleaved construct for freezing grids. Indeed, in Figure 1 Supplement 1, EYFP-Prestin is clearly not cleaved. Authors should comment this point. Additionally, the procedure of sample preparation is not explained in details, for instance they should comment the choice of the GDN micelles.

The reviewer is correct that we did not cleave YFP. We could not identify a density that corresponds to YFP in our structures presumably due to high mobility, and variable position. We have added text to the results section to clarify.

We have added explanations for digitonin and GDN as choices for detergent.

- In supplementary Figure 1 Supplement 2 authors could include also the Euler angle distribution of particles for the cryo-EM reconstruction to highlight distinct preferential orientations.

We have included the Euler angle distributions to Supplementary Figure 1, Supplement 3D.

Reviewer #3 (Remarks to the Author):

For context, this paper comes as part of a surge of 3 papers, reporting the eagerly awaited cryo-EM structure of the OHC motor protein prestin (SLC26A5).

Here, Butan et al. report a single structure of prestin from the gerbil.

Taken together, the structure (of reasonable, but not highest resolution) shows the basic features that have been expected, based on recent experimental structures of the homolog, SLC26A9, and a previous

modeling effort with prestin. These features confirmed by the new structure include the 7TMIR fold, global domain structure (core/gate), identity of the anion binding site and dimerization involving the domain-swapped STAS arrangement.

From there, the authors aim at understanding the mechanism by which prestin generates cellular motility ('piezoelectric behavior'). Although the manuscript lacks clarity in this respect, it appears that the tacit assumption is that electromotility results from transition between (2 or likely more) conformations that differ mechanically. Given that only a single state is described here (unlike in the 'companion' papers), no straightforward conclusions can be made as to the mechanism.

Sorry, the mechanism that we referred to is the intrinsic versus extrinsic voltage sensor mechanism in driving conformational changes in prestin. This concept is central to the piezoelectric-like behavior that prestin possesses. We believe that the structures of prestin in different conformations driven by voltage under normal conditions have yet to be determined, and thus we disagree with the Ge et al. paper on that. As we have indicated in our response above, a number of biophysical measures indicate that the voltage driven changes in conformation are likely quite different from the conformational states in the presence of salicylate and SO_4^{2-} . We have made the comments explicit in the discussion, and have modified the introduction to clarify these points.

However, the authors argue that they can exclude a transporter-like mechanism (for SLC26 transporters most likely an elevator-like movement of the core-domain) for electromotility. This is the most questionable aspect of the manuscript, presented in a difficult-to-follow manner, and conclusions rest on rather circumstantial evidence.

First, it is claimed that the electrical partial charges that contribute to the voltage sensitivity of electromotility are distributed across domains that are not expected to move in the elevator movement that has been postulated for SLC26 transporters. However, the actual transport dynamics have not been defined experimentally for any SLC26 transporter yet, hence the absence of movement in any domain is speculative.

Yes, in the absence of direct evidence, we must base our conclusions from models of the elevator mechanism (please see comments by Reviewer 1). These models, however, are based on experimentally determined structures in the inside open and outside open state of a number of transporters in the extended families.

Further, the conclusions rely mostly on previous findings from this group, where neutralizing 12 distinct residues individually (by mutation) was reported to reduce voltage dependence.

Yes, you are right that our conclusions and those of the other prestin structural papers rely heavily on our previous work.

Those effects should be interpreted very conservatively, as the overall sensor charge is well below 1 e and hence the contribution of each charged residue must be tiny, which is also true for the expected effects on NLC voltage dependence. Conspicuously, the sensor charges of the binding site mutations reported in the present manuscript all fall into the range considered to indicate reduced sensor charge in the previous paper - but here this is claimed to indicate 'normal unitary gating charge'.

This conclusion is a misleading. We correctly claim that in the present experiments control and experimental NLC are no different. In Fig. 3 B, top panel, the 2-state fit for z is 0.705 +/- 0.033, not

statistically different from the binding site mutations. Nor are our mutation results different from Ge et al.'s (2001) normal results. Sorry, we did not include the fit results and statistical comparisons in the Fig. 3 legend, which we now do. BTW, in our previous paper we also used statistical comparisons of fit results.

Also, charged residues may also impact on apparent voltage dependence by sculpting the local electrical field experienced by the mobile 'voltage sensor' charges .

We don't understand the relevance of this obvious point, should mobile charges exist. We do not think mobile charges contribute to NLC, nor does the original author of the extrinsic voltage sensor hypothesis, Dominik Oliver (Gorbunov et al., 2018). Moreover, the data from the Ge et al. paper showing movement of residues involved in chloride binding towards the cytoplasmic surface in prestin purified in the presence of SO_4^{2-} would also be against the extrinsic voltage sensor hypothesis. As we have cautioned, we do not believe that this state is the expanded state that corresponds to prestin's voltage driven expanded state for a number of reasons outlined above and in the discussion.

Also, I do not see how the presented binding site mutations exclude a contribution of the bound substrate anion to voltage sensor charge.

Residue mutations within a binding pocket are well known to alter binding (e.g., affinity) in many proteins. If normal binding of anions were required for sensor charge movement, then NLC would be altered, which it is not. The text has been changed to clarify the point.

It appears that the introduction of a glutamate into this site functionally replaces the extrinsic anion (but cannot be pushed out by salicylate) and the charge could move with the core domain in an elevator-like movement. Other mutations in the binding site are made, but how these experiments constrain the mechanism is unclear.

First, since there is no binding or unbinding, contributions of anion binding per se are not involved in NLC, as would be required with the extrinsic voltage sensor. Second, we view, based on a wealth of prior studies, anions as enabling voltage-dependency through an allosteric-like mechanism, the residue charge simply performs this requirement. Third, we previously showed that transport in prestin can be divorced from NLC generation. Fourth, mutation to any number of residues that partake in coordinating Cl^- binding have no effect on unitary charge movement, as would be expected were Cl^- to act as its voltage sensor.

It is then shown that the homolog, SLC26A9, lacks charge movement and hence electromotility (see concern below). The rationale for presenting these experiments is unclear.

The rationale is that structurally prestin and Slc26a9 are quite similar, so the absence of NLC in Slc26a9 can help understand prestin. Stated differently, if prestin's movements are akin to transporter movements, then Slc26a9, in which the majority of the charged residues in prestin responsible for NLC are conserved, should also show measurable NLC. That Slc26a9 does not is further evidence that transporter like movements don't underlie NLC. Moreover, other transporters in the same family in which many of the residues responsible for prestin's NLC are conserved, also do not show NLC (Oliver et al., 2001; Bai et al., 2009; Kuwabara et al., 2018). While the structures of these proteins are also likely similar to prestin, we are constrained by actual structural data to make comparisons with these. The text has been changed accordingly to clarify the points.

In general, it should be considered that electrogenicity and substrate charge vary across the SLC26 family, so the charge moved with the transport steps may differ despite a common transport mechanism expected from structural conservation.

This is speculation. Here we directly show that Slc26a9 does not exhibit NLC. The majority of transporters in the Slc26 family do not show NLC (Oliver et al., 2001; Bai et al., 2009; Kuwabara et al., 2018). The singular exceptions are prestin and pendrin (Kuwabara et al., 2018). There is no correlation that we can discern to the transported ions or their electrogenicity with NLC or its absence.

Specifically, SLC26A9 behaves like a chloride channel, and therefore may actually lack the transport transition. It may therefore not be the appropriate model for asking whether transport involves electromotility-like charge movement/NLC. On the contrary, the structurally most closely related SLC26 transporter (A5 from non-mammals) was reported to feature charge movement.

Slc26a9 has transporter like activity (Xu et al., 2005). Its channel like activity is likely therefore to be uncoupled transport, rather than channel activity (see comment by Reviewer 1). Finally, whether or not a family member exhibits channel like properties has no bearing on whether it can move residue charge during voltage perturbation.

Other points

1. In the experiments on SLC26A9 charge movement, the expression levels are questionable. According to other papers (e.g. Walter et al., 2019), chloride currents of the full-length protein expressed transiently can be on the order of nA. Here, no current at all is reported and the authors have to use SCN as an alternative substrate to see any functional expression at all. Maybe the expression is simply too low to allow detection of NLC.

This conclusion is wrong. We did report on membrane currents for Slc26a9 transfected cells in Fig. 4 and Figure 4, Supplement 1. The reviewer may have missed these figures and associated text, or perhaps views our Slc26a9 currents as small because we report them in nA/pF to correct for cell size. This is the proper way to evaluate current magnitude in transfected cells, where transfection efficiencies may vary. Walter et al. showed Cl⁻ currents spanning +/- 1.25 nA (their Fig. 4 E,F), with a range of 15 to 30 pF linear capacitance per cell. Taking 22.5 pF as average, this will convert to 0.057 nA/pF for their current data. Our maximum average current in the absence of SCN (that is, in the presence of Cl⁻) for untransfected cells is 0.01 nA/pF and 3.7 fold higher at 0.037 nA/pF for transfected cells. In the presence of SCN, it is an additional 3.5 fold higher at 0.13 nA/pF. Furthermore, we find that our transfection is successful based on membrane targeting. We believe our resolution for NLC measurement or gating charge movement is sufficient since we have measured real-time the delivery of femtofarad levels of prestin NLC to the membrane (Bian et al., 2013). The measurement approach in our program jClamp is a standard that many in the field use, including Ge et al., (2021). We are very confident that Slc26a9 presents no NLC.

2. It is argued that the state resolved here represents the contracted one. This seems plausible, considering prestin's voltage dependence. However, given the lack of any membrane-like environment together with the biophysically well documented impact of membrane composition and tension on prestin's state distribution, this conclusion appears less certain. The lack of membrane environment might force the protein into any native (or even non-native) state.

Reviewer 1 raised the same question, please see our response above. These are caveats that all structural studies must acknowledge. To the best of our knowledge, however, as we state in the paper, the likelihood that prestin is contracted at or near zero voltage is a reasonable assumption. Moreover, the structures in high chloride in detergent and in nanodiscs are the same and near identical to ours (Ge et al.,2021) making lipid influence on these particular structures less likely.

Bai J, Navaratnam D, Moeini-Naghani I, Li F-Y, Zhong S, Bian S, Santo-Sacchi J (2016) Prestin's Non Selective Currents are Mediated by an Auxillary Pathway that is Independent of its Transporter Pathway. ARO Midwinter Meeting Abstracts:438.

Bai JP, Surguchev A, Montoya S, Aronson PS, Santos-Sacchi J, Navaratnam D (2009) Prestin's anion transport and voltage-sensing capabilities are independent. Biophys J 96:3179-3186.

Bian S, Navaratnam D, Santos-Sacchi J (2013) Real time measures of prestin charge and fluorescence during plasma membrane trafficking reveal sub-tetrameric activity. PLoS One 8:e66078.

Chang YN, Geertsma ER (2017) The novel class of seven transmembrane segment inverted repeat carriers. Biol Chem 398:165-174.

Drew D, Boudker O (2016) Shared Molecular Mechanisms of Membrane Transporters. Annu Rev Biochem 85:543-572.

Ficici E, Faraldo-Gomez JD, Jennings ML, Forrest LR (2017) Asymmetry of inverted-topology repeats in the AE1 anion exchanger suggests an elevator-like mechanism. J Gen Physiol 149:1149-1164.

Gorbunov D, Hartmann J, Renigunta V, Oliver D (2018) A glutamate scan identifies an electrostatic switch for prestin activity. Midwinter Meeting Abstracts of the Association for Research in Otolaryngology.

Kuwabara MF, Wasano K, Takahashi S, Bodner J, Komori T, Uemura S, Zheng J, Shima T, Homma K (2018) The extracellular loop of pendrin and prestin modulates their voltage-sensing property. J Biol Chem 293:9970-9980.

Oliver D, He DZ, Klocker N, Ludwig J, Schulte U, Waldegger S, Ruppersberg JP, Dallos P, Fakler B (2001) Intracellular anions as the voltage sensor of prestin, the outer hair cell motor protein. Science 292:2340-2343.

- Rybalchenko V, Santos-Sacchi J (2008) Anion control of voltage sensing by the motor protein prestin in outer hair cells. *Biophys J* 95:4439-4447.
- Santos-Sacchi J, Navarrete E (2002) Voltage-dependent changes in specific membrane capacitance caused by prestin, the outer hair cell lateral membrane motor. *Pflugers Arch* 444:99-106.
- Santos-Sacchi J, Song L (2014) Chloride and Salicylate Influence Prestin-dependent Specific Membrane Capacitance. *Journal of Biological Chemistry* 289:10823-10830.
- Santos-Sacchi J, Song L, Zheng J, Nuttall AL (2006) Control of mammalian cochlear amplification by chloride anions. *J Neurosci* 26:3992-3998.
- Schanzler M, Fahlke C (2012) Anion transport by the cochlear motor protein prestin. *J Physiol* 590:259-272.
- Xu J, Henriksnas J, Barone S, Witte D, Shull GE, Forte JG, Holm L, Soleimani M (2005) SLC26A9 is expressed in gastric surface epithelial cells, mediates Cl⁻/HCO₃⁻ exchange, and is inhibited by NH₄⁺. *Am J Physiol Cell Physiol* 289:C493-505.

REVIEWER COMMENTS

Reviewer #1 (Remarks to the Author):

The response of the Authors and corresponding changes in the manuscript are satisfactory. I do not have any other comments and recommend the acceptance of the work.

Reviewer #3 (Remarks to the Author):

The manuscript clearly improved, as several issues are now presented in a more precise way.

With respect to my previous concerns, it is appreciated that the introduction now more precisely defines the mechanistic concept, in defining the extrinsic voltage sensor mechanism. Yet it should be noted that the present definition appears different from the original one (Oliver, 2001), where an unbound anion hopping through an access well is envisioned as carrying the sensor charge.

The current concept put forward in line 55 ('posits a transporter like movement with an arrested hemi-movement of Cl⁻ in the transporter cycle acting as its voltage sensor.') seems to more align with the 'hybrid voltage sensor' put forward by Bavi et al. 2021, based on their prestin structures.

Apart from a few minor issues (see below), two points (related to each other) remain unsatisfactorily addressed, and need to be resolved.

First, the level of evidence against the transporter-like extrinsic voltage sensor does not allow to exclude this mechanism with reasonable certainty.

Second, the conclusions drawn from the binding site mutagenesis are still flawed.

1. It is clear, that the 398E mutant excludes an anion movement relative to protein as the voltage sensor, because the fixed charge introduced by this mutation enables voltage sensing although fixing the position of the 'anion' relative the protein.

However, this result does certainly not exclude a transporter-like voltage sensor ('hybrid' or 'elevator-like' sensor) as described in the Introduction; in contrast, it strengthens the idea that occupation of the binding site is a prerequisite for voltage sensing/eM, and is entirely consistent with the 'hybrid' or 'elevator-like' sensor. This issue was also correctly brought up by reviewer 1.

Evidence brought forward against the 'transporter-like' hybrid (extrinsic) sensor model is essentially the localization of charged residues across the prestin structure, that were previously reported to affect sensor valence (l. 327 ff, Suppl. Fig. 6). But: (a) a movement of at least a subset of these charges as part of an intrinsic voltage sensor would not be incompatible with the transport-like model, in particular if core domain movement was accompanied by additional structural rearrangement including gate domain helices as suggested by Bavi et al.. (b) the role of some of these residues should be critically reviewed as their position in intracellular loops appears difficult to reconcile with an interaction with the electric field required for any contribution to voltage sensing.

There is nothing wrong (in my opinion) with a critical view to the 'transporter-like' extrinsic sensor model, but I strongly suggest to tone down the claims made here a bit.

Specifically this refers to the last sentence of the abstract. Besides being difficult to understand (what does 'any associated transport-like requirements' mean?), the statement is flawed, as occupation of the transporter substrate binding site strongly impacts on NLC, which already constitutes an intimate relation between transport mechanism and voltage sensing.

2. The second argument put forward against the 'transporter-like' extrinsic sensor model, is the presentation of binding site mutations that do not significantly affect apparent sensor valence z .

However, multiple binding site mutants mutant have been published (e.g. Bavi et al., 21; Schächinger et al., 2011, Gorbunov et al., 2014 ; and also from this group: Bai et al., 2017) that heavily change NLC, including complete loss of NLC. Therefore, the presentation of additional mutants that have no effect (on z at least) cannot prove irrelevance of anion binding. Mechanistically, and equally important, the present manuscript asserts that binding site alteration would be expected to alter apparent z for an extrinsic (read: hybrid/transporter-like) sensor model. This is wrong. Mutations of the residues that coordinate the anion are expected to reduce binding affinity, which alters occupancy but not sensor valence, because in this model the sensor can only move once the anion is bound (and valence is unaltered). Expected changes are rather a reduced charge movement/NLC due to lower occupancy and a shift in V_h (as state distribution is poised towards unoccupied states). This is exactly what has been shown for binding site mutations in several previous papers (mentioned above).

Whether this is also true for the mutations made in this study is unknown, because only z values are reported. If these mutants provide additional (and different) information than previous binding site mutants, the full NLC data of the mutants could (and should) be easily added to Figure 3, including representative or average traces, like in Fig.3A/B.

Specific point: Q94: this residue is not obviously involved in Cl coordination, it points away from the binding pocket. Also, I found no data that shows a major role in chloride binding in A9 (even though in A9 the homologous residue is oriented more towards the binding pocket than in prestin, see. Fig.3F).

Related: Figure legend 3: data for NLC fits need to be sorted: there are two WT control values that differ quite substantially, used for different statistical comparisons; one is attributed to a YFP fusion construct,

the other only to WT; then there are some data from HEK cells and some from CHO cells, I cannot get this sorted out unambiguously. Finally the numbers differ from those in the main text!

Other minor points

(rebuttal in italics)

3. *Specifically, SLC26A9 behaves like a chloride channel, and therefore may actually lack the transport transition. It may therefore not be the appropriate model for asking whether transport involves electromotility-like charge movement/NLC. On the contrary, the structurally most closely related SLC26 transporter (A5 from non-mammals) was reported to feature charge movement.*

Answer: Slc26a9 has transporter like activity (Xu et al., 2005). Its channel like activity is likely therefore to be uncoupled transport, rather than channel activity (see comment by Reviewer 1). Finally, whether or not a family member exhibits channel like properties has no bearing on whether it can move residue charge during voltage perturbation.

Although possible, it is not known at present. A9 is atypical in terms of transport mode within the SLC26 family and certainly differs from SLC26A5, which are coupled transporters. Therefore, absence of NLC in A9 may not be the strongest evidence against relation of electromotility to transport (elevator) dynamics. On the other hand, the SLC26 transporters most similar to prestin, non-mammalian A5, is shown to feature NLC. So together, the finding of absence of NLC may exclude electromotility in A9 - but does not help very much in clarifying (or excluding) a similarity between electromotility and elevator transport transitions.

4.

In the experiments on SLC26A9 charge movement, the expression levels are questionable. According to other papers (e.g. Walter et al., 2019), chloride currents of the full-length protein expressed transiently can be on the order of nA. Here, no current at all is reported and the authors have to use SCN as an alternative substrate to see any functional expression at all. Maybe the expression is simply too low to allow detection of NLC.

Answer: This conclusion is wrong. We did report on membrane currents for Slc26a9 transfected cells in Fig. 4 and Figure 4, Supplement 1. The reviewer may have missed these figures and associated text, or perhaps views our Slc26a9 currents as small because we report them in nA/pF to correct for cell size. This is the proper way to evaluate current magnitude in transfected cells, where transfection efficiencies

may vary. Walter et al. showed Cl⁻ currents spanning +/- 1.25 nA (their Fig. 4 E,F), with a range of 15 to 30 pF linear capacitance per cell. Taking 22.5 pF as average, this will convert to 0.057 nA/pF for their current data. Our maximum average current in the absence of SCN⁻ (that is, in the presence of Cl⁻) for untransfected cells is 0.01 nA/pF and 3.7 fold higher at 0.037 nA/pF for transfected cells. In the presence of SCN⁻, it is an additional 3.5 fold higher at 0.13 nA/pF. Furthermore, we find that our transfection is successful based on membrane targeting. We believe our resolution for NLC measurement or gating charge movement is sufficient since we have measured real-time the delivery of femtofarad levels of prestin NLC to the membrane (Bian et al., 2013). The measurement approach in our program jClamp is a standard that many in the field use, including Ge et al., (2021). We are very confident that Slc26a9 presents no NLC.

These numbers need to be given in the manuscript. My objection was based on the presentation in Fig.4, where currents in A9-transfected cells are clearly of the same size as those measured in the prestin-transfected cells. Even though prestin may mediate some anionic conductance (previous reports from the same group) comparable current amplitudes would not obviously qualify as strong expression. Mean currents of mock-transfected cells could easily be added to the same graph (Fig.4A) to indicate current level mediated by A9, avoiding misinterpretation.

More generally, in absence of even estimates for single-molecule conductance/rates, there is no robust way to compare expression levels based on these currents. The same can be noted for the fluorescence images, which are non-quantitative, at least in the way they are presented. I would suggest to at least try to give estimates for membrane fluorescence levels in comparison to prestin. Alternatively, Walter et al. have described an elegant way to boost membrane localization without compromising function by truncating the disordered part of the C terminus.

5.

It is argued that the state resolved here represents the contracted one. This seems plausible, considering prestin's voltage dependence. However, given the lack of any membrane-like environment together with the biophysically well documented impact of membrane composition and tension on prestin's state distribution, this conclusion appears less certain. The lack of membrane environment might force the protein into any native (or even non-native) state.

Answer: Reviewer 1 raised the same question, please see our response above. These are caveats that all structural studies must acknowledge. To the best of our knowledge, however, as we state in the paper, the likelihood that prestin is contracted at or near zero voltage is a reasonable assumption. Moreover, the structures in high chloride in detergent and in nanodiscs are the same and near identical to ours (Ge et al.,2021) making lipid influence on these particular structures less likely.

Agreed. It is a reasonable assumption. However, the wording (l.192) is still 'prestine must be in a contracted state'. Please revise.

6. Supplemental Fig.6: panels H, I, J should be labeled G, H, I

7. Figure legend to Supplemental Fig.6, l. 903: (D,E,F) should be (G,H,I)

8. l.223 , sentence is quite confusing: 'Additionally, we have shown z values based on fit results that also show no significant differences in the z values of WT and mutant prestin (Figure 3).'

Do you mean: 'These z values were not significantly different from WT (Figure 3)' ?

Response to Reviewers. (black reviewer comments; red our responses)

First, we want to thank the reviewers for critical suggestions that have made the manuscript much better.

Reviewer #1 (Remarks to the Author):

The response of the Authors and corresponding changes in the manuscript are satisfactory. I do not have any other comments and recommend the acceptance of the work.

Thank you.

Reviewer #3 (Remarks to the Author):

The manuscript clearly improved, as several issues are now presented in a more precise way.

With respect to my previous concerns, it is appreciated that the introduction now more precisely defines the mechanistic concept, in defining the extrinsic voltage sensor mechanism. Yet it should be noted that the present definition appears different from the original one (Oliver, 2001), where an unbound anion hopping through an access well is envisioned as carrying the sensor charge. The current concept put forward in line 55 ('posits a transporter like movement with an arrested hemi-movement of Cl⁻ in the transporter cycle acting as its voltage sensor.') seems to more align with the 'hybrid voltage sensor' put forward by Bavi et al. 2021, based on their prestin structures.

Let us try and explain more fully. The concept we describe is that of the original manuscript of Oliver et al. (2001) and their subsequent publications. Within the 2001 manuscript there is no figure depicting chloride ion movements in prestin.

Their model was subsequently better described in another of their publications (Gorbanov et al., 2014) which clearly shows transporter linkage (Fig. 7f), pasted below

We now include a citation (nearer to line 55) to the Gorbanov et al (2014) paper, as well as our more recent review, Santos-Sacchi, Navaratnam, Raphael and Oliver (2017), each already having been included in our paper at other text locations.

Apart from a few minor issues (see below), two points (related to each other) remain unsatisfactorily addressed, and need to be resolved.

First, the level of evidence against the transporter-like extrinsic voltage sensor does not allow to exclude this mechanism with reasonable certainty.

Second, the conclusions drawn from the binding site mutagenesis are still flawed.

1. It is clear, that the 398E mutant excludes an anion movement relative to protein as the voltage sensor, because the fixed charge introduced by this mutation enables voltage sensing although fixing the position of the 'anion' relative the protein.

Thank you.

However, this result does certainly not exclude a transporter-like voltage sensor ('hybrid' or 'elevator-like' sensor) as described in the Introduction; in contrast, it strengthens the idea that occupation of the binding site is a prerequisite for voltage sensing/eM, and is entirely consistent with the 'hybrid' or 'elevator-like' sensor. This issue was also correctly brought up by reviewer I.

Clearly the binding of chloride is critically important for NLC. As we noted, we have much data (cited by Ge et al. (2021) and Bavi et al. (2021)) indicating that the binding of chloride is allosteric-like, thus enabling voltage-dependence of prestin. The reviewer is incisive and correct that this is the central conundrum facing us – that is how does the binding of chloride then make the protein voltage sensitive? The structures of the many interesting mutants that Oliver (R399C for example) has since made will likely be most instructive in deciphering this conundrum.

Mutation alone cannot be considered the same as "occupation of the binding site", as with phospho-mimetic mutations for any number of reasons – ability to bind/ unbind, angles, charge etc.

A brief clarification that eM and NLC can be divorced, for example, by reducing turgor pressure in the OHC (Santos-Sacchi, 1991). A further quick note that salicylate does not "lock prestin into the expanded state" since substantial voltage-dependent eM persists as a linearized function of voltage (Kakehata and Santos-Sacchi, 1996; Wu and Santos-Sacchi, 1998)

Evidence brought forward against the 'transporter-like' hybrid (extrinsic) sensor model is essentially the localization of charged residues across the prestin structure, that were previously reported to affect sensor valence (l. 327 ff, Suppl. Fig. 6).

Both Ashmore's group and Fahlke's group along with us have confirmed transporter activity in prestin. Indeed, we additionally cite our previous work showing divorce of NLC and prestin transport, and report now results from additional pertinent mutations. We believe this is strong evidence.

But: (a) a movement of at least a subset of these charges as part of an intrinsic voltage sensor would not be incompatible with the transport-like model, in particular if core domain movement was accompanied by additional structural rearrangement including gate domain helices as suggested by Bavi et al.. (b) the role of some of these residues should be critically reviewed as their position in intracellular loops appears difficult to reconcile with an interaction with the electric field required for any contribution to voltage sensing.

Based on structure, the majority of the residues we identify are within the membrane. Furthermore, the static structures (that in the case of SO_4^{2-} may not relate to hyperpolarizing voltages) that are available now cannot ensure that any charged residue predicted to be outside the field will not enter the field upon conformational change. This is particularly the case since many of the residues that lie in the linker are in fact very close to the transmembrane segments. So, we think this exercise would not be worthwhile until we get some actual voltage-dependent structures, which we are currently working on.

There is nothing wrong (in my opinion) with a critical view to the 'transporter-like' extrinsic sensor model, but I strongly suggest to tone down the claims made here a bit.

Thank you. We should note that throughout the many years we have worked on this, we have agreed in print that there is no theoretical problem with intrinsic and extrinsic charge contributing to measurable displacement currents in prestin. We are simply trying to find which is dominant, and now we believe we have evidence to do so.

Specifically this refers to the last sentence of the abstract. Besides being difficult to understand (what does 'any associated transport-like requirements' mean?), the statement is flawed, as occupation of the transporter substrate binding site strongly impacts on NLC, which already constitutes an intimate relation between transport mechanism and voltage sensing.

Given the ambiguity in the understanding of the extrinsic model, we toned down the abstract and the discussion a bit now.

What we mean by transporter movement is the initial proposal by Oliver et al. (2001) that the hemi-transport of chloride constitutes the voltage sensor movement. What all structures of prestin have revealed is diametrically opposite to what was predicted. That is, chloride in its binding pocket in the contracted, inside open state faces farther away from the cytoplasm in the membrane than the expanded state. In Oliver's model the movement would be reversed in order to generate the measured positive gating charge that accompanies transition from the expanded to the contracted state.

What the elevator model predicts is an upward movement of the chloride binding site from the inside open to the outside open conformation – best exemplified in Lucy Forrest's delightful paper on AE1 in JGP (Ficici et al., 2017). What the Ge paper shows is an upward/downward movement of the binding site in two inside open structures, and not what is expected from the elevator model (Ge et al., 2021).

2. The second argument put forward against the 'transporter-like' extrinsic sensor model, is the presentation of binding site mutations that do not significantly affect apparent sensor valence z . However, multiple binding site mutants mutant have been published (e.g. Bavi et al., 21; Schächinger et al., 2011, Gorbunov et al., 2014 ; and also from this group: Bai et al., 2017) that heavily change NLC, including complete loss of NLC. Therefore, the presentation of additional mutants that have no effect (on z at least) cannot prove irrelevance of anion binding.

Mechanistically, and equally important, the present manuscript asserts that binding site alteration would be expected to alter apparent z for an extrinsic (read: hybrid/transporter-like) sensor model. This is wrong. Mutations of the residues that coordinate the anion are expected to reduce binding affinity, which alters occupancy but not sensor valence, because in this model the sensor can only move once the anion is bound (and valence is unaltered). Expected changes are rather a reduced charge movement/NLC due to lower occupancy and a shift in V_h (as state distribution is poised towards unoccupied states). This is exactly what has been shown for binding site mutations in several previous papers (mentioned above).

Whether this is also true for the mutations made in this study is unknown, because only z values are reported. If these mutants provide additional (and different) information than previous binding site mutants, the full NLC data of the mutants could (and should) be easily added to Figure 3, including representative or average traces, like in Fig.3A/B.

To clarify, when we responded to the question in our initial rebuttal that binding affinity, for example, may change upon pocket mutations, we did not say that change in affinity would alter z . The term z depends on the distance the charged ion (in the transporter case) moves perpendicular to the field. An alteration in the average position of the ion in the binding pocket (likely produced by mutations of the pocket residues) could in turn alter the displacement angle or displacement magnitude of charge movement during conformational change -- something that we did not see. Alterations in V_h are not necessarily reflected with binding site mutations, since a variety of mutations, even in the C terminus shift V_h (Bai et al., 2006); also membrane tension and temperature shifts V_h , without altering z (Okunade and Santos-Sacchi, 2013; Santos-Sacchi and Tan, 2020). Moreover, we find no relationship between V_h and z in single point mutations that affect z (Bai et al., 2009). Furthermore, "reduced charge movement/NLC" is most likely a measure of transfection efficiency and properly folded protein membrane targeting, not prestin function. We believe that z is the major parameter that should be affected by mutations of the binding site, since it measures movement of the protein with voltage. The reviewer is correct that a larger aim of these structural studies would be to decipher how V_h (or state occupancy) is affected by these many factors. The only point mutation where effects on V_h has been mechanistically established, that we know of, is the careful paper by Kazu Homma's group (Kuwabara et al., 2018), where they show charge screening effects of the extracellular loop connecting TMs 5 and 6 but not TMs 3 and 4. These residues on the extracellular loops like our mutations in the C terminus are not voltage sensor candidates.

Nevertheless, the V_h of all the mutants is included in the Figure text as suggested by the reviewer.

Specific point: Q94: this residue is not obviously involved in Cl⁻ coordination, it points away from the binding pocket. Also, I found no data that shows a major role in chloride binding in A9 (even though in A9 the homologous residue is oriented more towards the binding pocket than in prestin, see. Fig.3F).

Thanks, we agree. We initially posited that Q94 would coordinate Cl⁻ based on the similarity to Slc26a9. Based on the reviewers' comments, we reviewed the structure of prestin in the PDB and determine that the Q94 side chain points away from the Cl⁻ binding site and ion in the Ge et al. (2021) paper. We have therefore removed the data associated with the mutation from the manuscript.

While there may not be direct evidence of Cl⁻ binding in A9, it is a reasonable assumption since it transports chloride ((Walter et al., 2019); (Lohi et al., 2002; Xu et al., 2005; Loriol et al., 2008)).

Related: Figure legend 3: data for NLC fits need to be sorted: there are two WT control values that differ quite substantially, used for different statistical comparisons; one is attributed to a YFP fusion construct, the other only to WT;

We are sorry about the confusion. The two WT z means are 0.71 and 0.73 ($p=0.86$). These are not different. We collect data from control cells on each day of experimentation, and group those within the same timeframes. This accounts for the 0.02 mean difference in the two different controls. Both controls were prestin YFP. Experiments with S398E were done first. Experiments with the second set of mutants were done afterwards, about 3 weeks after S398E. Hence the two different control groups. We have added a clarification in methods.

then there are some data from HEK cells and some from CHO cells, I cannot get this sorted out unambiguously.

Again, we are sorry about the confusion. We compare mutation values in transfected CHO cells; we did not use our stable HEK cell line for mutational analysis, only for structural studies. We have added a clarification in methods and Figure legend.

Finally the numbers differ from those in the main text!

Thank you! You are right! This was due to rounding issues in the second decimal in the figure legend. We have corrected.

This is from the text: (Q94A 0.66 +/- 0.06, Q97A 0.65 +/- 0.04, P136T 0.64 +/- 0.05).

This is from the legend: (Q94A 0.65 +/- 0.06, Q97A 0.64 +/- 0.03, P136T 0.64 +/- 0.05).

The correct values are in the text and will be used throughout.

Other minor points
(rebuttal in italics)

3. Specifically, SLC26A9 behaves like a chloride channel, and therefore may actually lack the transport transition. It may therefore not be the appropriate model for asking whether transport involves electromotility-like charge movement/NLC. On the contrary, the structurally most closely related SLC26 transporter (A5 from non-mammals) was reported to feature charge movement.

Answer: Slc26a9 has transporter like activity (Xu et al., 2005). Its channel like activity is likely therefore to be uncoupled transport, rather than channel activity (see comment by Reviewer 1). Finally, whether or not a family member exhibits channel like properties has no bearing on whether it can move residue charge during voltage perturbation.

Although possible, it is not known at present.

The observation that it has transporter like activity was an interpretation of the authors (Xu et al., 2005).

A9 is atypical in terms of transport mode within the SLC26 family and certainly differs from SLC26A5, which are coupled transporters.

The paper by Xu et al (2005) that we cited showed coupled ($\text{Cl}^-/\text{HCO}_3^-$) transporter activity (Figure 4 in their paper).

Therefore, absence of NLC in A9 may not be the strongest evidence against relation of electromotility to transport (elevator) dynamics.

Yes, it may not be the strongest, however, it cannot be ignored.

On the other hand, the SLC26 transporters most similar to prestin, non-mammalian A5, is shown to feature NLC.

Yes, the reviewer is correct that it would be best to make the comparison with non-mammalian a5 and pendrin, that too has both NLC and transporter function! However, we do not have a structure of non-mammalian a5 or indeed any other Slc26 family member to make the comparison. This, along with the close alignment with its structure, were our reasons to compare prestin to Slc26a9, which has several structures.

So together, the finding of absence of NLC may exclude electromotility in A9 - but does not help very much in clarifying (or excluding) a similarity between electromotility and elevator transport transitions.

Our reasons for making the comparison with A9 is because our structure most resembles A9 and since we don't yet have other Slc26 structures. Our reason for pressing this point is that apart from pendrin, none of the other mammalian Slc26 family members have measurable NLC although they share many of the charged residues that are important for NLC.

4.

In the experiments on SLC26A9 charge movement, the expression levels are questionable. According to other papers (e.g. Walter et al., 2019), chloride currents of the full-length protein expressed transiently can be on the order of nA. Here, no current at all is reported and the authors have to use SCN as an alternative substrate to see any functional expression at all. Maybe the expression is simply too low to allow detection of NLC.

Answer: This conclusion is wrong. We did report on membrane currents for Slc26a9 transfected cells in Fig. 4 and Figure 4, Supplement 1. The reviewer may have missed these figures and associated text, or perhaps views our Slc26a9 currents as small because we report them in nA/pF to correct for cell size. This is the proper way to evaluate current magnitude in transfected cells, where transfection efficiencies may vary. Walter et al. showed Cl^- currents spanning +/- 1.25 nA (their Fig. 4 E,F), with a range of 15 to 30 pF linear capacitance per cell. Taking 22.5 pF as average, this will convert to 0.057 nA/pF for their current data. Our maximum average current in the absence of SCN- (that is, in the presence of Cl^-) for untransfected cells is 0.01 nA/pF and 3.7 fold higher at 0.037 nA/pF for transfected cells. In the presence of SCN-, it is an additional 3.5 fold higher at 0.13 nA/pF. Furthermore, we find that our transfection is successful based on membrane targeting. We believe our resolution for NLC measurement or gating charge movement is sufficient since we have measured real-time the delivery of femtofarad levels of prestin NLC to the membrane (Bian et al., 2013). The measurement approach in our program jClamp is a standard that many in the field use, including Ge et al., (2021). We are very confident that Slc26a9 presents no NLC.

These numbers need to be given in the manuscript. My objection was based on the presentation in Fig.4,

where currents in A9-transfected cells are clearly of the same size as those measured in the prestin-transfected cells. Even though prestin may mediate some anionic conductance (previous reports from the same group) comparable current amplitudes would not obviously qualify as strong expression. Mean currents of mock-transfected cells could easily be added to the same graph (Fig.4A) to indicate current level mediated by A9, avoiding misinterpretation.

Thanks. Agreed. Done.

More generally, in absence of even estimates for single-molecule conductance/rates, there is no robust way to compare expression levels based on these currents. The same can be noted for the fluorescence images, which are non-quantitative, at least in the way they are presented. I would suggest to at least try to give estimates for membrane fluorescence levels in comparison to prestin. Alternatively, Walter et al. have described an elegant way to boost membrane localization without compromising function by truncating the disordered part of the C terminus.

We are hesitant to investigate this deletion and use fluorescence as a quantitative marker for surface expression. Some background first. In particular, the paper by (Walter et al., 2019) is the main reason for our hesitation. In that paper, Walter et al made deletions excluding the IVS domain and noted, without quantification, increased surface expression based on fluorescence. They also noted an increase in currents with the truncation that they inadvertently attributed to increased surface expression. Rather, the subsequent extraordinary paper by Chi et al. (Chi et al., 2020) (where they determine the structure of A9 without truncation of the C terminal IVS) shows that C-terminal truncation results in a more than threefold increase in the single channel conductance due to an absent ball and chain blocking effect, similar to that in voltage gated ion-channels. Thus, the increased currents seen by (Walter et al., 2019) was most likely due to the absence of a current blocking effect from the C-terminus, since the increases in current were similar (more than 3 fold on single channel by Chi et al (Chi et al., 2020) and four fold in current by Walter et al (Walter et al., 2019).

Since we are not evaluating this truncation, currents should represent the native capabilities of the transporter and give a good estimate of membrane insertion (with similar values to Walter et al. (Walter et al., 2019), better than fluorescence. We considered, and refrained from making the truncations, since the transporter mechanisms of the truncations are not settled. We are using the standard ways to assess transfection used by (Walter et al., 2019) as well, and are satisfied with this approach.

5.

It is argued that the state resolved here represents the contracted one. This seems plausible, considering prestin's voltage dependence. However, given the lack of any membrane-like environment together with the biophysically well documented impact of membrane composition and tension on prestin's state distribution, this conclusion appears less certain. The lack of membrane environment might force the protein into any native (or even non-native) state.

Answer: Reviewer 1 raised the same question, please see our response above. These are caveats that all structural studies must acknowledge. To the best of our knowledge, however, as we state in the paper, the likelihood that prestin is contracted at or near zero voltage is a reasonable assumption. Moreover, the structures in high chloride in detergent and in nanodiscs are the same and near identical to ours (Ge et al., 2021) making lipid influence on these particular structures less likely.

Agreed. It is a reasonable assumption. However, the wording (l.192) is still 'prestins must be in a

contracted state'. Please revise.

Thank you. We have changed as pointed out.

6. Supplemental Fig.6: panels H, I, J should be labeled G, H, I

Thank you. We have changed as pointed out.

7. Figure legend to Supplemental Fig.6, l. 903: (D,E,F) should be (G,H,I)

Thank you. We have changed as pointed out.

8. l.223 , sentence is quite confusing: 'Additionally, we have shown z values based on fit results that also show no significant differences in the z values of WT and mutant prestin (Figure 3).'

Do you mean: 'These z values were not significantly different from WT (Figure 3)' ?

Thank you. We have changed as suggested.

- Bai J-P, Navaratnam D, Samaranayake H, Santos-Sacchi J (2006) En block C-terminal charge cluster reversals in prestin (SLC26A5): Effects on voltage-dependent electromechanical activity. *Neuroscience Letters* 404:270-275.
- Bai JP, Surguchev A, Montoya S, Aronson PS, Santos-Sacchi J, Navaratnam D (2009) Prestin's anion transport and voltage-sensing capabilities are independent. *Biophys J* 96:3179-3186.
- Chi X, Jin X, Chen Y, Lu X, Tu X, Li X, Zhang Y, Lei J, Huang J, Huang Z, Zhou Q, Pan X (2020) Structural insights into the gating mechanism of human SLC26A9 mediated by its C-terminal sequence. *Cell Discov* 6:55.
- Ficici E, Faraldo-Gomez JD, Jennings ML, Forrest LR (2017) Asymmetry of inverted-topology repeats in the AE1 anion exchanger suggests an elevator-like mechanism. *J Gen Physiol* 149:1149-1164.
- Ge J, Elferich J, Dehghani-Ghahnaviyeh S, Zhao Z, Meadows M, von Gersdorff H, Tajkhorshid E, Gouaux E (2021) Molecular mechanism of prestin electromotive signal amplification. *Cell* 184:4669-4679 e4613.
- Kakehata S, Santos-Sacchi J (1996) Effects of salicylate and lanthanides on outer hair cell motility and associated gating charge. *J Neurosci* 16:4881-4889.
- Kuwabara MF, Wasano K, Takahashi S, Bodner J, Komori T, Uemura S, Zheng J, Shima T, Homma K (2018) The extracellular loop of pendrin and prestin modulates their voltage-sensing property. *J Biol Chem* 293:9970-9980.
- Lohi H, Kujala M, Makela S, Lehtonen E, Kestila M, Saarialho-Kere U, Markovich D, Kere J (2002) Functional characterization of three novel tissue-specific anion exchangers SLC26A7, -A8, and -A9. *J Biol Chem* 277:14246-14254.
- Loriol C, Dulong S, Avella M, Gabillat N, Boulukos K, Borgese F, Ehrenfeld J (2008) Characterization of SLC26A9, facilitation of Cl(-) transport by bicarbonate. *Cell Physiol Biochem* 22:15-30.
- Okunade O, Santos-Sacchi J (2013) IR laser-induced perturbations of the voltage-dependent solute carrier protein SLC26a5. *Biophys J* 105:1822-1828.
- Santos-Sacchi J (1991) Reversible inhibition of voltage-dependent outer hair cell motility and capacitance. *J Neurosci* 11:3096-3110.
- Santos-Sacchi J, Tan W (2020) Complex nonlinear capacitance in outer hair cell macro-patches: effects of membrane tension. *Sci Rep* 10:6222.
- Walter JD, Sawicka M, Dutzler R (2019) Cryo-EM structures and functional characterization of murine Slc26a9 reveal mechanism of uncoupled chloride transport. *Elife* 8.
- Wu M, Santos-Sacchi J (1998) Effects of lipophilic ions on outer hair cell membrane capacitance and motility. *J Membr Biol* 166:111-118.

Xu J, Henriksnas J, Barone S, Witte D, Shull GE, Forte JG, Holm L, Soleimani M (2005) SLC26A9 is expressed in gastric surface epithelial cells, mediates Cl⁻/HCO₃⁻ exchange, and is inhibited by NH₄⁺. *Am J Physiol Cell Physiol* 289:C493-505.

REVIEWERS' COMMENTS

Reviewer #3 (Remarks to the Author):

The manuscript has been changed appropriately. I see no further problems.

There is just one small issue left, probably omitted accidentally:

'Fig.4, where currents in A9-transfected cells are clearly of the same size as those measured in the prestin-transfected cells. Even though prestin may mediate some anionic conductance (previous reports from the same group) comparable current amplitudes would not obviously qualify as strong expression. Mean currents of mock-transfected cells could easily be added to the same graph (Fig.4A) to indicate current level mediated by A9, avoiding misinterpretation.

Thanks. Agreed. Done.'

The amendment is missing in the revised version of the ms.

Response to Reviewers. (black reviewer comments; red our responses)

First, we want to thank the reviewers for critical suggestions that have made the manuscript much better.

Reviewer #3 (Remarks to the Author):

The manuscript has been changed appropriately. I see no further problems.

There is just one small issue left, probably omitted accidentally:

'Fig.4, where currents in A9-transfected cells are clearly of the same size as those measured in the prestin-transfected cells. Even though prestin may mediate some anionic conductance (previous reports from the same group) comparable current amplitudes would not obviously qualify as strong expression. Mean currents of mock-transfected cells could easily be added to the same graph (Fig.4A) to indicate current level mediated by A9, avoiding misinterpretation.

Thanks. Agreed. Done.'

The amendment is missing in the revised version of the ms.

The reviewer is correct that this was left out in omission. We apologize. We have now made the change to Figure 4 as suggested.